# Taming Text Alignment for Personalization: Disentangling Foreground Customization and Background Style in Diffusion Models for Personalized Image Generation

## Abstract

Personalized image generation focuses on synthesizing text-driven images conditioned on reference images aimed at specifying the details of the generated content. Typically, it involves two key subtasks: *image customization* and *style transfer*. Previous arts failed to align the generated image with the text prompts, specifying as misalignment with background for image customization and foreground for style transfer, owing to the dense image references that attend only to either the foreground or background, while overwhelming the sparse text prompts. To tackle the text-alignment for personalized image generation, in this paper, we propose a **D**ual-**R**eference **P**ersonalization diffusion model, dubbed **DRP-Diff**, for both image customization and style transfer to achieve the text alignment for personalized image generation, where the crux lies in disentangling the foreground customization and background style, so as to separately align each of them with text prompts during different stages of denoising process. To achieve the alignment between text prompt and customized reference focusing on background, throughout our *customized texture-disentangled matrix*, we concatenate the foreground texture of both the customized and style references as key in the cross-attention to reconstruct the query background of the denoised personalized image. To align the style reference focusing on foreground with text prompts, we serve the background of style reference via *style-disentangled matrix* as the key associated with its value to reconstruct the query foreground of the denoised personalized image. Each of the above two processes are conducted in the early and late stage of the whole denoising process guided by text prompts. To adaptively modulate the denoising timesteps between the early and late stages, the ratio of information entropy between the customized and style references is calculated. Extensive experiments validate the superiority of DRP-Diff over the state-of-the-art diffusion models for personalized image generation. *Our code can be accessed from the supplementary material package*.

## 1 Introduction

Personalized image generation focuses on synthesizing text-driven images conditioned on reference images, which specify the details of the generated content. The remarkable progress largely stems from the powerful generative capabilities of Denoising Diffusion Probabilistic Models (DDPMs) Ho et al. (2020), *e.g.*, Stable Diffusion Rombach et al. (2022); Esser et al. (2024); Podell et al. (2023). Typically, the personalized image generation involves two key subtasks, *i.e.*, *image customization* and *style transfer*. Given text prompts, image customization Xiao et al. (2025); Zeng et al. (2024); Pang et al. (2024); Gal et al. (2022); Ruiz et al. (2023); Kumari et al. (2023); Liu et al. (2025); Ma et al. (2024); Li et al. (2024; 2023) aims to generate images by substituting the foreground semantics in the text prompts with the foreground extracted from the reference images, while preserving the background of the text prompts. In contrast, style transfer Rout et al. (2025); Liu et al. (2023); Sohn et al. (2023); Zhu et al. (2025); Qi et al. (2024); Hertz et al. (2024); Lei et al. (2025); Gao et al. (2024); Xing et al. (2024); Wang et al. (2024b;a); Jeong et al. (2024) focuses on disentangling style (*background*) from reference images and maintaining the foreground alignment with text prompts.

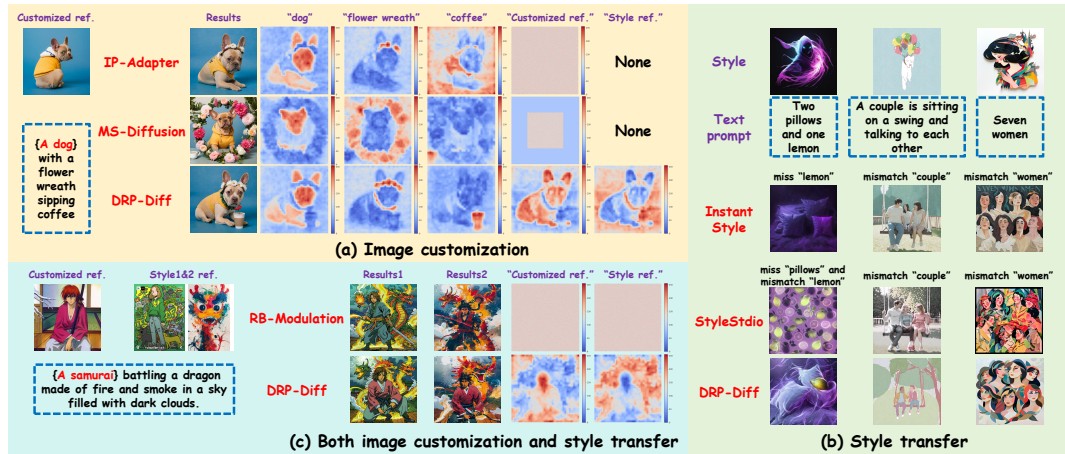

Figure 1: (a) Comparison of the generated results, as well as the attention score map of text prompts and image references, between our DRP-Diff and state-of-the-art IP-Adapter Ye et al. (2023) and MS-Diffusion Wang et al. (2025) for image customization. (b) Comparison of the generated results between our DRP-Diff and state-of-the-art InstantStyle Wang et al. (2024a) and StudyStdioLei et al. (2025) for style transfer. (c) Comparison of generated results and attention maps for both customized and style references between our DRP-Diff and RB-Modulation Rout et al. (2025).

However, prior arts have struggled to align the generated image with the text prompts within each individual subtask.

For image customization, prior methods Ye et al. (2023); Aiello et al. (2025); Huang et al. (2025b) tend to *emphasize dense image references of the foreground while overwhelming sparse text prompts of the background, especially for complex ones*, *e.g.*, a degraded "*flower wreath*" and the missing "*coffee*" in the IP-Adapter Ye et al. (2023) of Fig. 1(a). Building on this, layout-based methods Wang et al. (2025); Huang et al. (2025b); Zhang et al. (2024) exploit spatial layouts to explicitly constrain reference concepts to the foreground, while assigning background regions to text prompts. *This preserves the foreground identity, but the explicit separation of foreground and background in the layout disrupts overall consistency, leading to misalignment with (complex / long) text prompts.* As illustrated in the MS-Diffusion Wang et al. (2025) of Fig. 1(a), the generated spatial arrangement of "*flower wreath*" and "*coffee*" diverges from the prompt and instead resembles *"A flower wreath around a dog near a coffee"*.

From the perspective of style transfer, during the denoising process, especially in the early stage with the high-level noise, the foreground is reconstructed from sparse text prompts, *the dense style information intended for the background often leaks into the foreground, resulting in misalignment with the text prompts*, *e.g.*, the incorrect number of "*woman*" in the InstantStyle Wang et al. (2024a) of Fig. 1(b). To achieve the alignment between the foreground and the text prompts, the latest work StyleStudio Lei et al. (2025) applies style via instance normalization Huang & Belongie (2017) in the cross-attention layers and leverages layout to preserve foreground in the self-attention layers. Nevertheless, self-attention score maps mainly highlight only the most salient regions, implying that the extracted layout can align only with simple text prompts (*i.e.*, "*a dog*") rather than the multi-objects ("*two pillows and one lemon*" in the StyleStudio Lei et al. (2025) of Fig. 1(b)) text prompts, thereby *preventing it from handling the complex text prompts*.

So far, both image customization and style transfer exhibit suboptimal alignment with the text prompts. In particular, image customization primarily suffers from background misalignment, whereas style transfer is prone to foreground misalignment. This misalignment arises from the dense image references that attend only to either the foreground or background, while overwhelming the sparse text prompts. To this end, we propose a **D**ual-**R**eference **P**ersonalization diffusion model, named **DRP-Diff** (see Fig. 2(a)). Given text prompt, unlike the existing personalized image generation with single reference image as input, DPR-Diff simultaneously handle both customized and style reference images to achieve the goal. Concretely, our basic idea is to disentangle the foreground customization and background style, so as to separately align each of them with text prompts during the different stages of denoising process. Specifically:

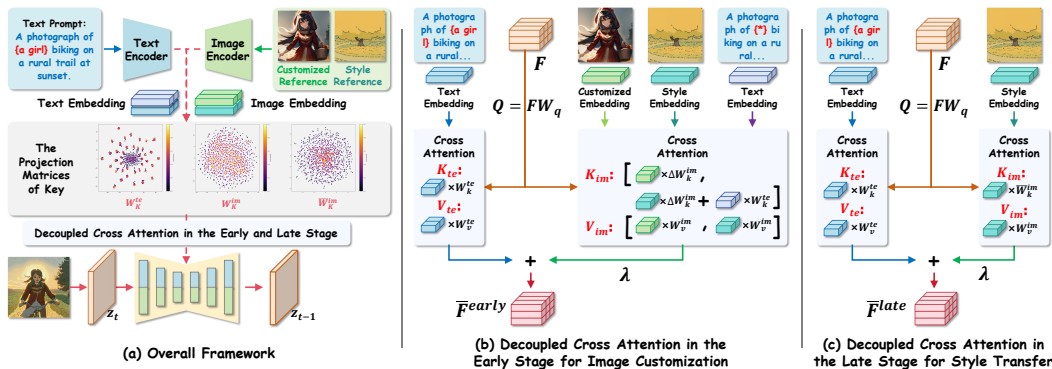

Figure 2: Illustration of our proposed DRP-Diff pipeline (a); (b) to achieve the alignment between text prompt and customized reference focusing on background, throughout our *customized texture-disentangled matrix* $\Delta W_k^{im}$ (Eq. (6)), we concatenate the foreground texture of both the customized $C_{cu}$ and style reference $C_{st}$ as the key $K_{im}$ to reconstruct the query background of denoised latent image feature $F$ (Sec. 2.3); and (c) to align the style reference focusing on foreground with text prompts, we serve the background of style reference $C_{st}$ via *style-disentangled matrix* $\overline{W}_k^{im}$ (Eq. (5)) as the key $K_{im}$ to reconstruct the query foreground of denoised latent image feature $F$ (Sec. 2.4). Each of the above two processes are conducted in the *early* and *late* stage of the whole denoising process guided by text prompts as illustrated in Eq. (1).

- We reformulate the key projection matrices in the decoupled cross-attention layer via singular value decomposition (SVD), yielding a *style-disentangled matrix* for the background style and a *customized texture-disentangled matrix* for the foreground customization. (Sec. 2.2).
- To achieve the alignment between text prompt and customized reference focusing on background, throughout our customized texture-disentangled matrix, we concatenate the foreground texture of both the customized reference with that of the style reference as key in the cross-attention, to obtain complementary information for reconstructing the query background of the denoised personalized image, as illustrated in Fig. 2(b) and DRP-Diff of Fig. 1(a). To align the style reference focusing on foreground with text prompts, we serve background of style reference via style-disentangled matrix as the key to reconstruct query foreground of the denoised personalized image, as illustrated in Fig. 2(c) and DRP-Diff of Fig. 1(c).
- Fig. 2(b) and 2(c) are conducted in the *early* and *late* stage of the whole denoising process guided by text prompts, owing to the high-level noise in the *early* stage to benefit the background reconstruction for customized reference, while prefer foreground reconstruction for style reference under the low-level noise in the *late* stage. To adaptively modulate the denoising timesteps between the early and late stages, the ratio of information entropy between the customized and style references is calculated (Sec. 2.4). Extensive experiments validate the superiority of DRP-Diff over the state-of-the-art diffusion models for personalized image generation.

## 2 METHODOLOGY

Central to our proposed DRP-Diff lies in the following aspects: 1) how to disentangle the foreground and background of the both image references(Sec. 2.2); 2) how to achieve the alignment between text prompt and customized reference focusing on background in the early stage (Sec. 2.3); 3) how to align the style reference focusing on foreground with text prompts in the late stage, along with the adaptively modulation for the timesteps within the denoising process between early and late stages (Sec. 2.4); before shedding light on DRP-Diff, we elaborate the preliminary regarding decoupled cross-attention mechanism, which plays a pivotal role of personalized image generation. (***Due to page limitation, more details about the algorithm in Sec.B of the Appendix***)

### 2.1 PRELIMINARY: DECOUPLED CROSS-ATTENTION MECHANISM IN DENOISING PROCESS

Often, personalized image generation methods adopt a decoupled cross-attention mechanism to integrate text prompt $C_{te} \in \mathbb{R}^{s_{te} \times d_{te}}$, the reference image $C_{im} \in \mathbb{R}^{s_{im} \times d_{im}}$ into every cross-attention

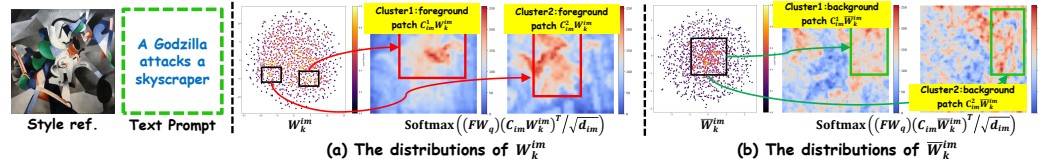

Figure 3: (a) we employ t-SNE to visualize the distributions on two clusters in the first column of $W_k^{im}$ (Eq. (4)) to highlight the foreground, while the attention score maps of Eq. (2) in the second and third columns indicate that the denoised query matrix $FW_q$ attends more on two foreground patches $C_{im}^1 W_k^{im}$ and $C_{im}^2 W_k^{im}$. (b) the style-disentangled matrix $\overline{W}_k^{im}$ in Eq. (5) highlight the background, leading $FW_q$ to attend more on two background patches $C_{im}^1 \overline{W}_k^{im}$ and $C_{im}^2 \overline{W}_k^{im}$.

layer of the denoising network where $s_{te}$ and $s_{im}$ denote the token number of the text prompts and image patches, such that $d_{te}$ and $d_{im}$ are the dimension for token and patch representation. Given the denoised latent image feature $F \in \mathbb{R}^{(h \times w) \times d}$ at the $l$-th layer of denoising UNet architecture denotes as $\epsilon_\theta(F, t, C_{te}, C_{im})$ at the $t$-th timestep, which served as the query to be reconstructed by the key of the text prompts via the cross-attention layers for the UNet architecture, as formulated below:

$$\overline{F}_{te} = Atten(Q, K_{te}, V_{te}) = \text{Softmax}\left(\frac{QK_{te}^T}{\sqrt{d_{te}}}\right) V_{te}, \tag{1}$$

where $Q \in \mathbb{R}^{(h \times w) \times d}$ denotes the query matrix, such that $Q = FW_q$ with $W_q \in \mathbb{R}^{d \times d}$ being as trainable linear projection matrix. Likewise, $K_{te} \in \mathbb{R}^{s_{te} \times d}$ and $V_{te} \in \mathbb{R}^{s_{te} \times d}$ denote the key and value matrices, such that $K = C_{te} W_k^{te}$ and $V = C_{te} W_v^{te}$ with $W_k^{te} \in \mathbb{R}^{d_{te} \times d}$ and $W_v^{te} \in \mathbb{R}^{d_{te} \times d}$ being as trainable linear projection matrices. Meanwhile, $F$ is fed into another cross-attention module to interact with the reference image embeddings below:

$$\overline{F}_{im} = Atten(Q, K_{im}, V_{im}) = \text{Softmax}\left(\frac{QK_{im}^T}{\sqrt{d_{im}}}\right) V_{im}, \tag{2}$$

where $K_{im} \in \mathbb{R}^{s_{im} \times d}$ and $V_{im} \in \mathbb{R}^{s_{im} \times d}$ denote the key and value matrices, such that $K = C_{im} W_k^{im}$ and $V = C_{im} W_v^{im}$ with $W_k^{im} \in \mathbb{R}^{d_{im} \times d}$ and $W_v^{im} \in \mathbb{R}^{d_{im} \times d}$ being as trainable linear projection matrices. The final output of the decoupled cross-attention $\overline{F}$ is then obtained by:

$$\overline{F} = \overline{F}_{te} + \lambda \overline{F}_{im}, \tag{3}$$

where $\lambda \in [0, 1]$ is weight factor, and the model becomes the original text-to-image diffusion model if $\lambda = 0$. The predicted noise, *i.e.*, output of $\epsilon_\theta(F, t, C_{te}, C_{im})$, is then subtracted from $F$ to update the latent feature, such denoising process iterates until $t = 0$, yielding the final personalized image.

## 2.2 Disentangling the Style and the Customized Texture of the image reference

Given text prompts, the major bottleneck for personalized image generation with dual reference lies in the entanglement (as shown in RB-Modulation Rout et al. (2025) of Fig. 1(c)). To resolve that, our basic idea is to disentangle the foreground and background of the both image reference, which paves the way to achieve the text alignment with them both.

**Disentangling the Style from the Background of the Image Reference.** Following Eq. (2), to disentangle the foreground and background of the image reference within the decoupled cross-attention layer, we visualize $W_k^{im}$ to analyze the regions where its contributions are predominantly concentrated, as illustrated in the first column of Fig. 3(a), where $W_k^{im}$ exhibits a **multi-cluster density distribution** with each cluster representing a foreground patch, so that the attention score among different patches well encode their correlations, resulting into the concentration on reconstructing the foreground of the denoised latent image feature $Q$ via $V_{im}$ for noise prediction. The predicted noise is then subtracted from the noisy latent to update the latent feature, and this process iterates until $t = 0$, yielding the final personalized image. This further weakens the background representation of the image reference, imposing the challenge in style disentanglement. To mitigate this issue, we perform singular value decomposition (SVD) on $W_k^{im}$:

$$W_k^{im} = U\Sigma V^T, \quad U \in \mathbb{R}^{d_{im} \times d_{im}}, \ \Sigma = \text{diag}(\sigma_1, \sigma_2, \dots, \sigma_{d_{im}}) \in \mathbb{R}^{d_{im} \times d_{im}}, \ V \in \mathbb{R}^{d \times d_{im}}, \tag{4}$$

(a) The Generated results of $W_k^{im}$ | (b) The Generated results of $\Delta W_k^{im}$

Figure 4: We investigate the image customization results using (a) $W_k^{im}$ (Eq. (4)), and (b) the customized texture-disentangled matrix $\Delta W_k^{im}$ (Eq. (6)), which can generate the personalized images to align with the text prompts while preserving the foreground texture of the customized reference.

where the singular values $\sigma_i$ in $\Sigma$ reflect the relative importance of each component: **larger $\sigma_i$ tend to correspond to the foreground, whereas smaller $\sigma_i$ are more likely associated with the background**. To suppress the dominant foreground components and enhance the relatively weak background components, we invert the singular values and construct the style-disentangled matrix $\overline{W}_k^{im}$ as follows:

$$\overline{W}_k^{im} = U\Sigma^{-1}V^T = \left(V\Sigma^{-1}U^T\right)^T = \left(W_k^{im+}\right)^T, \quad \Sigma^{-1} = \text{diag}\left(\tfrac{1}{\sigma_1}, \tfrac{1}{\sigma_2}, \ldots, \tfrac{1}{\sigma_{d_{im}}}\right), \quad \sigma_i \neq 0, \quad (5)$$

where $W_k^{im+} \in \mathbb{R}^{d \times d_{im}}$ denotes the **Moore–Penrose pseudo-inverse**[1] of $W_k^{im}$. Eq. (5) provides an intuitive means of disentangling the background from the foreground as the style in the cross-attention layer. As illustrated in the first column of Fig. 3(b), $\overline{W}_k^{im}$ exhibits a **centralized high-density cluster distribution**, where a single cluster pattern indicates that the representation primarily captures cross-patch similarities corresponding to shared background information. The second and third columns of Fig. 3(b) further demonstrates that the background extracted from the image reference predominantly contributes to reconstructing background of the denoised latent image feature. ***Due to page limitation, we provide more intuitions for the rationale between $W_k^{im}$ and $\overline{W}_k^{im}$ in Sec.D of the Appendix.***

**Disentangling the Customized Texture from the Foreground of the Image Reference.** Although $W_k^{im}$ in Eq. (4) can emphasizes the foreground more than the background, however, as illustrated in Fig. 4(a), it always generate the images that are basically similar to the reference image with no relevance upon the text prompts, due to the strong correlations as seen in Fig. 3(a). To remedy such issue, we weaken such foreground patch correlations by subtracting the background $\overline{W}_k^{im}$ Eq. (5) as follows:

$$\Delta W_k^{im} = W_k^{im} - \overline{W}_k^{im} = U\,\text{diag}(\tilde{\delta}_i)V^T, \quad \tilde{\delta}_i\big|_{\sigma_i \neq 0} = \sigma_i - \frac{1}{\sigma_i}, \tag{6}$$

where $\Delta W_k^{im}$ donates the customized texture-disentangled matrix which delivers the followings: by subtracting the inverted singular values ($\sigma_i - 1/\sigma_i$), background components help reduce the global correlations of the foreground without significantly altering regions with strong foreground semantics. As illustrated in Fig. 4(b), the $\Delta W_k^{im}$ can generate images that are aligned with the text prompts while preserving the texture of the image reference.

With the disentanglement of the foreground and background, we are ready to achieve the text alignment for both image customization and style transfer. To this end, we conduct the *early* and *late* stage of the whole denoising process guided by text prompts, owing to the high-level noise in the *early* stage to benefit the background reconstruction for customized reference, while prefer foreground reconstruction for style reference under the low-level noise in the *late* stage. (***Due to page limitation, more details about the relation between denoising process and text prompts in Sec.C of the Appendix***)

## 2.3 Early Stage of the Customized Texture-disentangled denoising process for image customization

To achieve the alignment between text prompt and customized reference focusing on background, throughout the $\Delta W_k^{im}$, we concatenate the foreground texture of the customized reference with

---

[1]The Moore–Penrose pseudo-inverse of $W_k^{im}$ is *parameter-dependent* and *independent of the input data*. Therefore, it can be pre-computed *once during the model loading stage*, saving computation cost during inference. Experimental verification of this is provided in Sec. 3.2 and Table.1.

that of the style reference in the cross-attention, to obtain complementary information for reconstructing the background of the denoised personalized image, the overall pipeline is illustrated in Fig. 2(b). Formally, we initialize the denoising process with random Gaussian noise $z_T$, together with the text prompts $C_{te}$ describing both foreground and background, the background-specific prompts $\overline{C}_{te} \in \mathbb{R}^{s_{te} \times d_{te}}$, the customized reference $C_{cu} \in \mathbb{R}^{s_{im} \times d_{im}}$, and the style reference $C_{st} \in \mathbb{R}^{s_{im} \times d_{im}}$, as inputs to the denoising network $\epsilon_\theta(F, t, C_{te}, \overline{C}_{te}, C_{cu}, C_{st})$ to generate the denoised results for the image customization. To *decouple the foreground and background in personalized image generation*, we reformulate Eq. (2) as follows:

$$\overline{F}_{im}^{early} = \text{Softmax}\left( \frac{(FW_q)[\overbrace{C_{cu}\Delta W_k^{im}}^{\substack{\text{Decoupled}\\\text{Foreground Term}}}; \overbrace{C_{st}\Delta W_k^{im}}^{\substack{\text{Decoupled}\\\text{Background Term}}} + \overbrace{\hat{C}_{te}W_k^{te}}^{\substack{\text{Auxiliary}\\\text{Position Term}}}]^T}{\sqrt{d_{im}}} \right)[C_{cu}W_v^{im}; C_{st}W_v^{im}], \qquad (7)$$

where $\hat{C}_{te} \in \mathbb{R}^{s_{im} \times d_{te}}$ denotes the embeddings of the text prompts for the background, so as to match the number of the patches $s_{im}$ for style reference $C_{st}$, we set each row $\hat{C}_{te}[i, ;] = \frac{1}{s_{te}} \sum_{i=1}^{s_{te}} \overline{C}_{te}[i, :]$ $(i = 1, 2, \ldots, s_{te})$ to be the average of the text token embedding vector, to guide style reference $C_{st}$ to be *globally* semantical to *background*. Following the above intuition, the final output of the decoupled cross-attention in the early stage is $\overline{F}^{early} = \overline{F}_{te} + \lambda \overline{F}_{im}^{early}$.

Concretely, Eq. (7) consists of three components: 1) the **decoupled foreground term** $C_{cu}\Delta W_k^{im}$ leverages the foreground texture of the customized reference to reconstruct the foreground of $F$. 2) the **decoupled background term** $C_{st}\Delta W_k^{im}$ leverages the foreground texture of the style reference to provide complementary information for reconstructing the background of $F$, by expanding the key and value sets in the cross-attention layers, thereby facilitating the query $FW_q$ to attend to more regions $C_{st}W_v^{im}$, so as to achieve alignment between the text prompt and the customized reference with a focus on the background. 3) the **auxiliary position term** $\hat{C}_{te}W_k^{te}$ attends more to background of style reference $C_{st}\Delta W_k^{im}$, thereby alleviating the inference with the customized reference.

## 2.4 LATE STAGE OF THE STYLE-DISENTANGLED DENOISING PROCESS FOR STYLE TRANSFER

To align the style reference focusing on foreground with text prompts, we serve the background of style reference via style-disentangled matrices as the key associated with its value to reconstruct the foreground of the denoised personalized image. Consequently, in the late stage, given the text prompts $C_{te}$ and the style reference $C_{st}$ as inputs to the denoising network $\epsilon_\theta(F, t, C_{te}, C_{st})$, we incorporate the background information from the style reference via the style-disentangled matrix $\overline{W}_k^{im}$ in Eq. (5), to produce the denoised result for style transfer, which can be formulated as follows:

$$\overline{F}_{im}^{late} = \text{Softmax}\left( \frac{(FW_q)\left(C_{st}\overline{W}_k^{im}\right)^T}{\sqrt{d_{im}}} \right)\left(C_{st}W_v^{im}\right), \quad \overline{F}^{late} = \overline{F}_{te} + \lambda \overline{F}_{im}^{late}, \qquad (8)$$

where $\overline{F}^{late}$ is the final output of the decoupled cross-attention in the late stage, the above modules is shown in Fig. 2(c). Based on the above, the crux is how to modulate the early and late stage? To adaptively *modulate* the denoising timesteps between the early and late stages, we calculate the threshold $\Delta_{cs}$ as follows:

$$\Delta_{cs} = \frac{H_{cu}}{H_{st}} = \frac{-\sum_{i=1}^{d_{im}} \text{Softmax}(C_{cu})_i \log(\text{Softmax}(C_{cu})_i + \epsilon)}{-\sum_{i=1}^{d_{im}} \text{Softmax}(C_{st})_i \log(\text{Softmax}(C_{st})_i + \epsilon)}, \qquad (9)$$

where $H_{cu}$ and $H_{st}$ are the information entropies of the customized and style references, and $\epsilon$ is a small constant ensuring numerical stability. A larger ratio $\Delta_{cs}$ indicates that the customized reference contains more meaningful semantics than the style reference, hence more timesteps in the *early* stage for image customization and fewer timesteps in the *late* stage for style transfer are required. For the timesteps of the denoising process from $T$ to $0$, as per $\Delta_{cs}$ in Eq. (9), we adaptively define the timesteps interval within $t \in [\frac{1}{1+\Delta_{cs}}T, T]$ that covers the early stage of the denoising process, while the late stage as $t \in [0, \frac{1}{1+\Delta_{cs}}T)$. (***See more results in Table. 3 and Sec. E of the Appendix***)

Table 1: Quantitative comparisons between **DRP-Diff** and other diffusion-based personalization models on several datasets. ▦ and ▦ denote the best and second-best results, respectively. **DRP-Diff** consistently achieves the best performance.

(a) Compared results with state-of-the-arts for both image customization and style transfer on DreamBooth, TI benchmark, and DreamBooth++.

| | DreamBooth and TI benchmark | | | | | | | DreamBooth++ | | | | | | |
| | IP-Adapter | SSR-Encoder | MIP-Adapter | Dreamcache | PatchDPO | MS-Diffusion | | IP-Adapter | SSR-Encoder | MIP-Adapter | Dreamcache | PatchDPO | MS-Diffusion | |
| | Arxiv'23 | CVPR'24 | AAAI'25 | CVPR'25 | CVPR'25 | ICLR'25 | | Arxiv'23 | CVPR'24 | AAAI'25 | CVPR'25 | CVPR'25 | ICLR'25 | |
| | {BASELINE} + STYLESSP (CVPR'25) | | | | | | DRP-DIFF | {BASELINE} + STYLESSP (CVPR'25) | | | | | | DRP-DIFF |
| CLIP-T ↑ | 0.2651 | 0.2664 | 0.2590 | 0.2490 | 0.2588 | 0.2937 | 0.3196 | 0.2632 | 0.2649 | 0.2578 | 0.2632 | 0.2595 | 0.2876 | 0.3108 |
| CLIP-TC ↑ | 0.4543 | 0.4783 | 0.4504 | 0.4241 | 0.4531 | 0.4451 | 0.4875 | 0.4785 | 0.4675 | 0.4727 | 0.4468 | 0.4744 | 0.4629 | 0.5122 |
| CLIP-TS ↑ | 0.4426 | 0.4443 | 0.4470 | 0.4482 | 0.4451 | 0.4631 | 0.4612 | 0.4368 | 0.4370 | 0.4348 | 0.4453 | 0.4375 | 0.4534 | 0.4558 |
| CLIP-CS ↑ | 0.6636 | 0.6592 | 0.6670 | 0.6426 | 0.6685 | 0.6460 | 0.6714 | 0.6724 | 0.6592 | 0.6685 | 0.6440 | 0.6714 | 0.6465 | 0.6821 |
| DINO-CS ↑ | 0.4088 | 0.3864 | 0.4036 | 0.3588 | 0.3988 | 0.3709 | 0.4089 | 0.4267 | 0.3901 | 0.4213 | 0.3721 | 0.4204 | 0.3734 | 0.4262 |
| IR×10 ↑ | -10.5572 | -10.4534 | -12.1468 | -11.7708 | -12.5709 | 1.4488 | 6.9093 | -11.1308 | -10.4540 | -12.0027 | -10.4016 | -11.6970 | -0.4119 | 1.9510 |
| HPS×10 ↑ | 2.0392 | 2.0346 | 1.9178 | 1.8243 | 1.9353 | 2.2801 | 2.5960 | 1.9782 | 1.9959 | 1.8972 | 1.8881 | 1.9054 | 2.1933 | 2.5260 |
| Time(s) ↓ | 9.61 | 7.34 | 12.16 | 8.48 | 9.42 | 10.99 | 5.66[10.54] | 9.61 | 7.34 | 12.16 | 8.48 | 9.42 | 10.99 | 5.66[10.54] |
| VRAM(GB) ↓ | 4.63 | 1.59 | 4.53 | 1.51 | 4.63 | 4.56 | 4.55 | 4.63 | 1.59 | 4.53 | 1.51 | 4.63 | 4.56 | 4.55 |

(b) Compared results with state-of-the-arts with methods for image customization on DreamBooth.

| Metric | Real Images | TI | DreamBooth | Custom-Diff | IP-Adapter | SSR-Encoder | BLIP-Diff | MS-Diff | Subject-Diff | JeDi | DRP-Diff |
| | - | Arxiv'22 | CVPR'23 | CVPR'23 | Arxiv'23 | CVPR'24 | NeurIPS'23 | ICLR'25 | SIGGRAPH 2024 | CVPR'24 | |
| DINO | 0.774 | 0.569 | 0.668 | 0.643 | 0.608 | 0.612 | 0.594 | 0.671 | 0.711 | 0.679 | 0.707 |
| CLIP-I | 0.885 | 0.780 | 0.803 | 0.790 | 0.809 | 0.821 | 0.779 | 0.792 | 0.787 | 0.814 | 0.833 |
| CLIP-T | N/A | 0.255 | 0.305 | 0.305 | 0.274 | 0.308 | 0.300 | 0.321 | 0.293 | 0.293 | 0.323 |
| Avg. | N/A | 0.535 | 0.592 | 0.579 | 0.564 | 0.580 | 0.558 | 0.595 | 0.597 | 0.595 | 0.621 |

# 3 EXPERIMENTS

## 3.1 IMPLEMENTATION DETAILS

We evaluate **DRP-Diff** on combined datasets covering image customization, style transfer, and text-to-image (T2I) generation. For small-scale both customization and style transfer, we use **Dream-Bench** Ruiz et al. (2023) and **TI benchmark** Gal et al. (2022) as customization references with **IS-Plus** Wang et al. (2024b) as style references; for large-scale settings, we adopt **DreamBench++** Peng et al. (2025) and **StyleBench** Gao et al. (2024); and for style transfer, we combine **IS-Plus** and **StyleBench** with prompts from **T2I-CompBench++** Huang et al. (2025a). We provide 9 evaluation metrics: (i) **Text and Image Alignment** via **CLIP**[2] Radford et al. (2021) and **DINO** Caron et al. (2021); (ii) **Image Quality** via **Image Reward (IR)** Xu et al. (2023) and **HPS** Wu et al. (2023); and (iii) **Efficiency** via inference time (**Times**) and GPU memory (**VRAM**). All experiments are implemented in PyTorch on a RTX 4090 GPU using **IP-Adapter** Ye et al. (2023) with **SDXL** Podell et al. (2023), 30 steps, and official hyper-parameters for fair comparison. (***Due to page limitation, more discussions about the implementation details in Sec. F of the Appendix***)

## 3.2 COMPARISON WITH STATE-OF-THE-ARTS

To validate the superiority of **DRP-Diff**, we perform a thorough comparison against state-of-the-art diffusion models for either image customization or style transfer. For fair comparison, we also compare with both image customization and style transfer. For the first stage of image customization, we consider two groups of representative methods. The first group includes **IP-Adapter** Ye et al. (2023), **Dreamcache** Aiello et al. (2025), and **PatchDPO** Huang et al. (2025b), while these approaches emphasize dense image references of the foreground while overwhelming sparse text prompts of the background, especially for complex ones. The second group consists of **SSR-Encoder** Zhang et al. (2024), **MIP-Adapter** Huang et al. (2025b), and **MS-Diffusion** Wang et al. (2025), resorting to layout constraints to integrate customized references. This preserves the foreground identity, but the explicit separation of foreground and background in the layout disrupts overall consistency, leading to misalignment with (complex / long) text prompts. Building on the above, we adopt the latest

---

[2]Since CLIP is trained on real image–text (T) pairs, it is particularly sensitive to image style, which may distort text–image similarity. To alleviate this, we propose CLIP-TS by measuring similarity with style reference (S), thereby reducing the style influence . In addition, the generated background aligned with text prompts can interfere with the customized reference (C); hence we propose CLIP-TC, which incorporates text prompts to suppress such interference. We also provide CLIP-T (image–text similarity) and CLIP-CS (image–image reference similarity) for comparison.

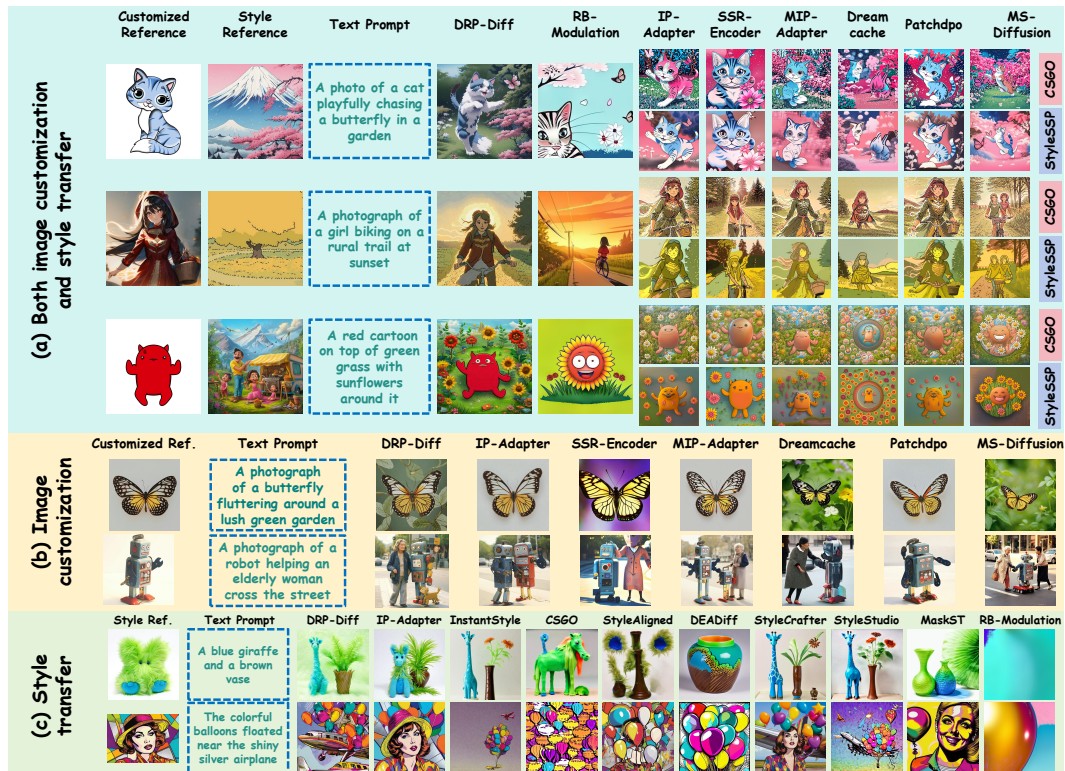

Figure 5: Comparison of (a) both image customization and style transfer, (b) image customization, and (c) style transfer with state-of-the-arts. **DRP-Diff** delivers superior results compared with existing approaches.

image-driven style transfer methods for the second stage of the style transfer, including **CSGO** Xing et al. (2024), **StyleID** Chung et al. (2024), and **StyleSSP** Xu et al. (2025). Last but not least, **RB-Modulation** Rout et al. (2025) can simultaneously take the customized and style references as inputs, but it cannot decouple the foreground and background regions.

**Quantitative and Qualitative analysis.** The quantitative results in Table. 1 summarize our findings below: **DRP-Diff** enjoys larger CLIP-T, CLIP-TC, CLIP-TS, CLIP-CS, DINO-CS, IR and HPS than the competitors. Notably, [**MS-Diffusion** Wang et al. (2025) + **StyleSSP** Xu et al. (2025)] remains the large performance margins at most 8.82% for CLIP-T, 9.53% for CLIP-TC, 3.93% for CLIP-CS 10.25% for DINO-CS and 13.86% for HPS) compared to **DRP-Diff** in Table. 1(a), the results verifies the intuition in Sec. 1 – *the dense image references that attend only to either the foreground or background, while overwhelming the sparse text prompts.* We also illustrate more personalized image generation results in Fig. 5(a). We further analyze the inference time of our **DRP-Diff** with two parts: **5.66s** for the denoising process and **4.88s** for computing Eq. (5) and Eq. (6). This demonstrates superior efficiency compared with state-of-the-art methods. Moreover, since Eq. (5) and Eq. (6) are parameter-dependent and unrelated to the input data, they can be pre-computed during the model loading stage, thereby saving this computation cost during inference. Our **DRP-Diff** also achieves competitive results in VRAM consumption. Specifically, **SSR-Encoder** adopts SD v1.5 and **Dreamcache** is designed for lightweight usage, so both consume less VRAM. However, compared with other SDXL-based models, our **DRP-Diff** achieves the lowest VRAM usage because it completes both tasks within a single denoising process. We also provide the result compared with the image customization methods in Table. 1(b) to exhibit the advantages of **DRP-Diff** by disentangling the foreground customization and background style. Fig. 5(b) and (c) provide the generation results of the image customization and style transfer, validating the superiority of **DRP-Diff**. (*Due to page limitation, see compared results in Sec. G, H and I of the Appendix for style transfer, along with more compared results on joint image customization and style transfer. See more generation results about different T2I models in Sec. J of the Appendix*)

Table 2: Ablation studies are conducted to evaluate three case for manipulating singular values: $\sigma_{\text{top-}n\%=0}$, which removes the largest $n\%$ of singular values by setting them to zero; $\sigma e^{-\sigma}$, a reweighting scheme that suppresses larger singular values while retaining smaller ones; and ours ($\frac{1}{\sigma}$), which inversely scales singular values. These strategies are examined to assess their effectiveness in suppressing foreground components and enhancing background components.

| Method | CLIP-T-Mean↑ | CLIP-T-Std↓ | CLIP-I-Mean↑ | CLIP-I-Std↓ |
|---|---|---|---|---|
| $\sigma_{\text{top-}10\%=0}$ | 0.2469 (-0.68%) | 0.0352 (+13.5%) | 0.6450 (+0.45%) | 0.0867 (+8.36%) |
| $\sigma_{\text{top-}20\%=0}$ | 0.2496 (+0.40%) | 0.0344 (+11.0%) | 0.6433 (+0.19%) | 0.0806 (+0.62%) |
| $\sigma_{\text{top-}30\%=0}$ | 0.2500 (+0.56%) | 0.0332 (+7.10%) | 0.6401 (-0.31%) | 0.0842 (+5.11%) |
| $\sigma_{\text{top-}40\%=0}$ | 0.2495 (+0.36%) | 0.0334 (+7.74%) | 0.6382 (-0.30%) | 0.0826 (+3.11%) |
| $\sigma_{\text{top-}50\%=0}$ | 0.2499 (+0.52%) | 0.0322 (+3.87%) | 0.6416 (-0.08%) | 0.0868 (+8.36%) |
| $\sigma e^{-\sigma}$ | 0.2498 (+0.49%) | 0.0330 (+6.45%) | 0.6401 (-0.31%) | 0.0883 (+10.2%) |
| Ours ($\frac{1}{\sigma}$) | 0.2486 | 0.0310 | 0.6421 | 0.0801 |

## 3.3 Ablation Studies

### Discussion on the Different Strategy to Disentangle Foreground and Background via Manipulating Singular Values

We conduct additional ablation studies to investigate different strategies for suppressing foreground components and enhancing background components in the singular values. Specifically, we consider all cases for the hard suppression approach $\sigma_{\text{top-}n\%=0}$, which denotes the top n% of singular values to zero, and the soft suppression approach $\sigma e^{-\sigma}$, which represents a reweighting scheme that attenuates larger singular values while preserving smaller ones. Table. 2 reports the results for all cases: overall, none of the strategies exhibits a noticeable advantage in extracting background information, as reflected by the similar **CLIP-T** and **CLIP-I** mean scores. The major differences lie in the fluctuations across images, measured by the **CLIP-T** and **CLIP-I** standard deviations. Our method achieves the most stable background disentanglement across diverse image scenarios.

- Setting different proportions of the top 10–50% of singular values to zero does not yield a clear improvement in performance. This is because the hard partitioning of foreground and background components makes **it difficult to adaptively determine the proportion of foreground and background for each image**, leading to limited performance gains and high variability across images.

- The soft suppression approach $\sigma e^{-\sigma}$ fails to effectively disentangle background for each image. While **it adaptively suppresses foreground components, it does not explicitly enhance background components**. As a result, the contribution of background components in the cross-attention remains limited, leading to incomplete background disentanglement.

- Orthogonally, our reciprocal operation $1/\sigma$ **not only weakens the dominant foreground components but also actively amplifies the smaller singular values corresponding to background**. This balanced adjustment ensures both suppression of foreground and enhancement of background, which results in more stable and consistent background disentanglement across images, confirming the intuition in Sec. 2.2.

**Discussion on the Early Stage for Image Customization.** To further ablate the early stage of **DRP-Diff**, named **DRP-Diff\***, on **DreamBench++** with three cases: **Case A**: discarding the style reference $C_{st}\Delta W_k^{im}$ in Eq. (7); **Case B**: removing the customized texture-disentangled matrix $\Delta W_k^{im}$ in Eq. (6); and **Case C**: removing the auxiliary term $\hat{C}_{te}W_k^{te}$ in Eq. (7). Table. 3(a) reports the results for all three cases, which suggests that **DRP-Diff\*** outperforms **Case A** when *concatenating the foreground texture of the customized reference with that of the style reference in the cross-attention*, which in line with Sec. 2.3. **Case B** exhibits a suboptimal performance degradation, confirming that $\Delta W_k^{im}$ *can generate images that are aligned with the text prompts while preserving the texture of the image reference*, confirming Eq. (6) in Sec. 2.2. Our **DRP-Diff\*** maintains achieves the large performance gain over **Case C**, implying $\hat{C}_{te}W_k^{te}$ *tends to alleviate the interference of the style reference with the customized reference*, as discussed in Sec. 2.3. Fig. 6 can also validate that.

**Discussion on the Late Stage for Style Transfer.** We further conduct ablation studies on the variant denoising process for style transfer on **DreamBench++** with three variants: **Case D**: removing

Table 3: Ablation studies about the influence of different early and late stage on generating the personalized images. (a) Case A (decoupled background term $C_{st}\Delta W_k^{im}$ in Eq. (7)), Case B ($\Delta W_k^{im}$ in Eq. (6)) and Case C (the auxiliary position term $\hat{C}_{te}W_k^{te}$ in Eq. (7)); (b) Case D ($\overline{W}_k^{im}$ in Eq. (5) and $\Delta W_k^{im}$ in Eq. (6)), Case E ($\overline{W}_k^{im}$ in Eq. (5)) and Case F ($\Delta_{cs}$ in Eq. (9)) for the late stage.

|  | Method | CLIP-T↑ | CLIP-TC↑ | Avg(CLIP-T,TC)↑ |  | Method | CLIP-T↑ | CLIP-TS↑ | Avg(CLIP-T,TS)↑ |
|---|---|---|---|---|---|---|---|---|---|
| (a) Early stage | Case A | 0.3191 | 0.5615 | 0.4403 | (b) Late stage | Case D | 0.3267 | 0.4397 | 0.3832 |
|  | Case B | 0.3354 | 0.5601 | 0.4478 |  | Case E | 0.3232 | 0.4514 | 0.3873 |
|  | Case C | 0.3179 | 0.5732 | 0.4456 |  | Case F | 0.3229 | 0.4690 | 0.3960 |
|  | **DRP-Diff*** | **0.3362** | **0.5742** | **0.4552** |  | **DRP-Diff** | **0.3359** | **0.4709** | **0.4034** |

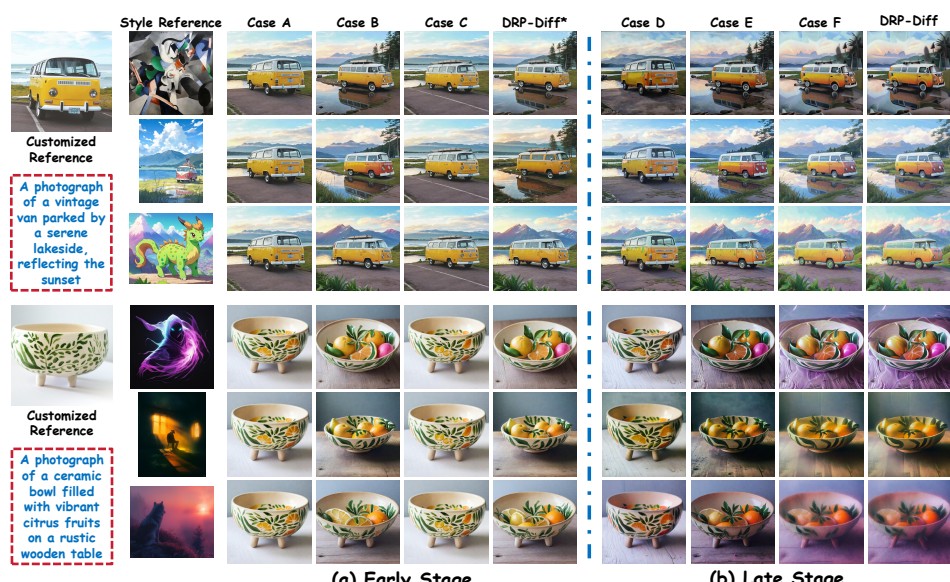

Figure 6: Ablation studies about the influence of different cross-attention of early and late stage on generating the personalized images. (a) Case A (decoupled background term $C_{st}\Delta W_k^{im}$ in Eq. (7)), Case B ($\Delta W_k^{im}$ in Eq. (6)) and Case C (the auxiliary position term $\hat{C}_{te}W_k^{te}$ in Eq. (7)); (b) Case D ($\overline{W}_k^{im}$ in Eq. (5) and $\Delta W_k^{im}$ in Eq. (6)), Case E ($\overline{W}_k^{im}$ in Eq. (5)) and Case F ($\Delta_{cs}$ in Eq. (9)) for the late stage.

the style-disentangled matrix $\overline{W}_k^{im}$ in Eq. (5) and the customization-disentangled matrix $\Delta W_k^{im}$ in Eq. (6); **Case E**: removing the style-disentangled matrix $\overline{W}_k^{im}$ in Eq. (5); **Case F**: removing $\Delta_{cs}$ in Eq. (9) and setting half of the denoising process as the early stage and the other as the late stage. Table. 3(b) reports that our **DRP-Diff** maintains large performance gains compared to **Case D**, which indicates that **DRP-Diff** outperforms by *disentangle the foreground and background of the references* (Sec. 1). **DRP-Diff** outperforms **Case E**, confirming that *our style-disentangled matrices can enhance the relatively weak background components*; **Case F** exhibits a suboptimal performance degradation, confirming that *the larger ratio in Eq. (9) indicates that the customized reference contains more meaningful semantics than the style reference, hence more timesteps in the early stage and fewer timesteps in the late stage*, confirming the intuition of Sec. 2.4.

## 4 CONCLUSION

In this paper, we propose a dual-reference personalization diffusion model, named DRP-Diff, for both image customization and style transfer to achieve the text alignment for personalized image generation by disentangling the foreground customization and background style, in the early and late stage of denoising process, together with an adaptive division modulation strategy between two stages. Extensive experiments validate the superiority of DRP-Diff over the state-of-the-art diffusion models for personalized image generation.

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

# A APPENDIX

Due to page limitation of the mainbody, as indicated by our submission, the appendix offers further more quantitative results and qualitative results with higher resolution, which are summarized below:

- The algorithm of joint image customization and style transfer, as mentioned in Sec. 2 of the mainbody (Sec. B)

- The relation between denoising process and text prompts, as mentioned in Sec. 2.2 of the mainbody (Sec. C)

- More intuitions for the rationale between $W_k^{im}$ and $\overline{W}_k^{im}$, as mentioned in Sec. 2.2 of the mainbody (Sec. D)

- More discussion on the impact of ratio $\Delta_{cs}$ in Eq.(9) for early and late denoising stage, as mentioned in Sec. 2.4 of the mainbody. (Sec. E)

- More details about the implementation details, as mentioned in Sec. 3.1 of the mainbody. (Sec. F)

- Additional quantitative results and qualitative analysis with *higher resolution* for the comparison with state-of-the-arts in style transfer, as mentioned in Sec. 3.2 of the mainbody. (Sec. G)

- Additional generate results about complexness for inter-object relationship for style transfer, as mentioned in Sec. 3.2 of the mainbody. (Sec. H)

- Additional quantitative results and qualitative analysis with *higher resolution* for the comparison with state-of-the-arts in both image customization and style transfer, as mentioned in Sec. 3.2 of the mainbody. (Sec. I)

- More generation results of different T2I models, as mentioned in Sec. 3.2 of the mainbody. (Sec. J)

- We provide the code for reproducibility. (Sec. K)

- Ethics Statement (Sec. L)

- Reproducibility Statement (Sec. M)

- The Use of Large Language Models (Sec. N)

# B THE ALGORITHM OF JOINT IMAGE CUSTOMIZATION AND STYLE TRANSFER

Due to the page limitation, we summarize the process of joint image customization and style transfer and present the detailed steps in Algorithm 1 as follow:

---

**Algorithm 1:** DRP-Diff : Joint Image Customization and Style Transfer

---

**Input:** the random Gaussian noise $z_T$, the text prompts $C_{te}$ including foreground and background, the text prompts of background $\overline{C}_{te}$, the customized reference $C_{cu}$, the style reference $C_{st}$, the denoising network $\epsilon_\theta$, the timestep $T$ and the decoder $\mathcal{D}$

**Output:** The personalized generated image $I_0$

1 Calculating the threshold $\Delta_{cs}$ via Eq.(9);

2 **for** $t = T, \ldots, \frac{1}{1+\Delta_{cs}}T$ **do**

3 $\quad$ Computing $z_{t-1}$ from $z_t$ is performed in the decoupled cross-attention (Eq.(1) and Eq.(7)) of $\epsilon_\theta$, conditioned on $C_{te}, \overline{C}_{te}, C_{cu}, C_{st}$, following Eq.(6).

4 **end**

5 **for** $t = \frac{1}{1+\Delta_{cs}}T, \ldots, 0$ **do**

6 $\quad$ Computing $z_{t-1}$ from $z_t$ is performed in the decoupled cross-attention (Eq.(1) and Eq.(8)) of $\epsilon_\theta$, conditioned on $C_{te}, C_{st}$, following Eq.(5).

7 **end**

8 **return** Output the personalized generated image $I_0 = \mathcal{D}(z_0)$

---

Table 4: The high-and-low frequency bands across different timesteps during the denoising process.

| Timesteps | Mean Gray | Mean HSV-V | Mean Luma | Avg Gradient | Variance | LBP Variance |
|---|---|---|---|---|---|---|
| **0** | 120.9 | 136.0 | 120.9 | 59.01 | 3955.0 | 4.45 |
| 5 | 120.8 | 134.6 | 120.8 | 75.07 | 4098.3 | 3.71 |
| 10 | 120.7 | 133.5 | 120.7 | 101.58 | 4378.9 | 3.25 |
| 15 | 120.7 | 133.7 | 120.7 | 133.26 | 4790.8 | 3.07 |
| 20 | 120.9 | 135.1 | 120.9 | 164.23 | 5274.8 | 3.07 |
| 25 | 121.2 | 137.3 | 121.2 | 190.49 | 5743.9 | 3.15 |
| 30 | 121.4 | 139.2 | 121.4 | 209.84 | 6099.5 | 3.25 |
| 35 | 121.1 | 140.0 | 121.1 | 222.03 | 6291.1 | 3.32 |
| 40 | 120.5 | 140.1 | 120.5 | 228.75 | 6363.9 | 3.35 |
| 45 | 119.9 | 139.8 | 119.9 | 232.19 | 6394.8 | 3.38 |
| 50 | 119.5 | 139.5 | 119.5 | 233.88 | 6419.5 | 3.40 |

To tackle the text-alignment for personalized image generation, in this paper, we propose a **D**ual-**R**eference **P**ersonalization diffusion model, dubbed **DRP-Diff**, for both image customization and style transfer to achieve the text alignment for personalized image generation. To achieve the alignment between text prompt and customized reference focusing on background, throughout our customized texture-disentangled matrix, we concatenate the foreground texture of both the customized reference with that of the style reference as key in the cross-attention, to obtain complementary information for reconstructing the query background of the denoised personalized image, as illustrated in Fig. 2(b) of the mainbody. To align the style reference focusing on foreground with text prompts, we serve background of style reference via style-disentangled matrix as the key to reconstruct query foreground of the denoised personalized image, as illustrated in Fig. 2(c) of the mainbody. Fig. 2(b) and 2(c) of the mainbody are conducted in the *early* and *late* stage of the whole denoising process guided by text prompts, owing to the high-level noise in the *early* stage to benefit the background reconstruction for customized reference, while prefer foreground reconstruction for style reference under the low-level noise in the *late* stage. To adaptively modulate the denoising timesteps between the early and late stages, the ratio of information entropy between the customized and style references is calculated.

## C  THE RELATION BETWEEN DENOISING PROCESS AND TEXT PROMPTS

As mentioned in Sec. 2.2 of the mainbody, *owing to the high-level noise in the early stage to benefit the background reconstruction for customized reference, while prefer foreground reconstruction for style reference under the low-level noise in the late stage.* We further claim that **during the denoising process, images recover low-frequency band first, followed by high-frequency details**. **Text prompt primarily corresponds to low-frequency band**, so in the early denoising stage with high-level noise, it is most effective for generating the overall content and background structure. **Foreground is already guided by the image reference, so the text prompt mainly contributes to generating the background, ensuring semantic alignment with the text**. In the late stage, as the noise level decreases, while the foreground and background are largely established, style references can be used to refine foreground details, thereby enhancing high-frequency information.

To validate the claims via frequency analysis, we experimentally measure the changes in both low- and high-frequency bands at different timesteps during the denoising process. The results are reported in Table. 4:

- **Low-frequency information.** We adopt *Mean Gray*, *Mean HSV-V*, and *Mean Luma*, which primarily capture global brightness and luminance structure. These metrics quantify the stability of the low-frequency bands throughout the denoising process.
- **High-frequency information.** We use *Average Gradient*, *Variance*, and *LBP Variance*, which are sensitive to edge sharpness, local contrast, and texture complexity. These metrics reflect the recovery of high-frequency details.

The low-frequency metrics remain nearly unchanged from the beginning (e.g., **Mean Gray varies only slightly from 119.5 to 120.9**). In contrast, the high-frequency metrics increase steadily throughout the reverse process (e.g., **Avg Gradient rises from 59.01 to 233.88**), confirming that

Table 5: The examples of text prompts for both foreground and background, foreground, and background to discussion intuitions for the rationale between $W_k^{im}$ and $\overline{W}_k^{im}$.

| ID | Both foreground and background | Foreground | Background |
|----|-------------------------------|-----------|-----------|
| 1 | A snow leopard is leaping across a fractured frozen expanse. | a snow leopard | the fractured frozen expanse |
| 2 | A giant squid is drifting in a radiant deep chasm. | a giant squid | the radiant deep chasm |
| 3 | A phoenix is glowing over an age-worn conflict field. | the phoenix | the age-worn conflict field |
| 4 | A red-crowned crane is gliding through a crimson-lit horizon. | a red-crowned crane | the crimson horizon |
| 5 | A polar bear is sitting beside a fractured icy dome-like form. | a polar bear | the fractured icy dome |
| 6 | A chameleon is blending into a glare-soaked luminous corridor. | a chameleon | the luminous corridor |
| 7 | A mountain gorilla is climbing an overrun observational platform. | a mountain gorilla | the overrun platform |
| 8 | A humpback whale is singing beneath drifting frozen masses. | a humpback whale | the drifting frozen masses |
| 9 | A komodo dragon is sunbathing on dark volcanic granules. | a komodo dragon | the volcanic granules |
| 10 | A peregrine falcon is landing on a ruined high outpost. | a peregrine falcon | the ruined high outpost |

Table 6: Quantitative comparisons of foreground–background disentanglement across different images via $W_k^{im}$ and $\overline{W}_k^{im}$.

| Method | Foreground | | | Background | | |
|--------|-----------|--------|--------|-----------|--------|--------|
| | CLIP-T↑ | CLIP-I↑ | DINO-I↑ | CLIP-T↑ | CLIP-I↑ | DINO-I↑ |
| Large singular values | 0.3115 | 0.8228 | 0.5774 | 0.2303 | 0.6157 | 0.1353 |
| Small singular values | 0.2629 | 0.7090 | 0.2591 | 0.2486 | 0.6421 | 0.1966 |

high-frequency details are refined gradually during denoising. This is consistent with **Sec. 1 of the mainbody**.

# D  MORE INTUITIONS FOR THE RATIONALE BETWEEN $W_k^{im}$ AND $\overline{W}_k^{im}$

## D.1  DISCUSSION ON THE INTUITIONS OF THE DIFFERENT SINGULAR VALUES

As indicated in Sec. 2.2 of the mainbody, we provide more intuitions for the rationale between $W_k^{im}$ and $\overline{W}_k^{im}$. To verify the intuition that **larger $\sigma_i$ tends to correspond to the foreground, whereas smaller $\sigma_i$ is more likely associated with the background**, as shown in Table. 5, We further made up more results as shown in Table. 5, where we utilize the text prompts for both the foreground and background to generate their corresponding images using a text-to-image model. These generated images are then used as image references in our **DRP-Diff** framework, where we disentangle the foreground via the large singular values and the background via the small singular values. For quantitative evaluation, we compute the similarity between the disentangled images and the text prompts for foreground or background in the Table. 5 via the CLIP model. Further, we generate images from text prompts for the foreground or background using a text-to-image model, and then measure the similarity between the disentangled images and these generated images. Both CLIP and DINO models are employed to assess the image-level similarity, providing a comprehensive evaluation of the consistency of the foreground–background disentanglement.

The quantitative results summarize our findings below: as illustrated in the left of the Table. 6, the large singular values is able to disentangle more foreground information than the small singular values, as evidenced by the higher scores in CLIP-T, CLIP-I, and DINO-I, whereas the small singular values is more effective at disentangling background information, as shown in the right of the Table. 6, confirming the intuition in **Sec. 2.2 of the mainbody**.

## D.2  DISCUSSION ON THE SINGULAR VALUES IN THE DIFFERENT LAYERS

we further analysis the singular values in the different cross-attention layers based on the singular value decomposition (SVD) on $W_k^{im}$ (Eq.(4) of the mainbody) and $\overline{W}_k^{im}$ (Eq.(5) of the mainbody) . For the singular value matrices obtained from different layers, we compute the proportion of the top n% singular values relative to the sum of all singular values, named $\sum \sigma_{\text{top-n\%}}$. We also take the reciprocal of each singular value and perform the same calculation. In both cases, the proportions are evaluated for $n \in 2, 5, 10, 20$. Table. 7 indicates that: under the same spatial resolution, the

Table 7: Summation of the top-$k\%$ singular values and their reciprocal counterparts across different cross-attention layers.

| Layer Name | Res. | $\sum \sigma_{\text{top-2\%}}$ | $\sum \sigma_{\text{top-5\%}}$ | $\sum \sigma_{\text{top-10\%}}$ | $\sum \sigma_{\text{top-20\%}}$ | $\sum \frac{1}{\sigma}_{\text{top-2\%}}$ | $\sum \frac{1}{\sigma}_{\text{top-5\%}}$ | $\sum \frac{1}{\sigma}_{\text{top-10\%}}$ | $\sum \frac{1}{\sigma}_{\text{top-20\%}}$ |
|---|---|---|---|---|---|---|---|---|---|
| down_blocks.1.0.0 | 64×64 | 0.127 | 0.259 | 0.408 | 0.606 | 0.520 | 0.724 | 0.849 | 0.934 |
| down_blocks.1.1.1 | 64×64 | 0.147 | 0.279 | 0.431 | 0.635 | 0.562 | 0.756 | 0.869 | 0.944 |
| down_blocks.2.0.0 | 32×32 | 0.197 | 0.360 | 0.544 | 0.763 | 0.525 | 0.715 | 0.838 | 0.927 |
| down_blocks.2.1.9 | 32×32 | 0.228 | 0.392 | 0.562 | 0.751 | 0.552 | 0.733 | 0.849 | 0.932 |
| mid_block.0.0 | 32×32 | 0.208 | 0.384 | 0.573 | 0.790 | 0.533 | 0.721 | 0.842 | 0.929 |
| mid_block.0.9 | 32×32 | 0.234 | 0.381 | 0.536 | 0.719 | 0.563 | 0.739 | 0.853 | 0.934 |
| up_blocks.0.0.0 | 32×32 | 0.179 | 0.344 | 0.532 | 0.755 | 0.518 | 0.710 | 0.835 | 0.926 |
| up_blocks.0.2.9 | 32×32 | 0.285 | 0.489 | 0.652 | 0.783 | 0.556 | 0.735 | 0.850 | 0.933 |
| up_blocks.1.0.0 | 64×64 | 0.130 | 0.275 | 0.437 | 0.641 | 0.537 | 0.739 | 0.859 | 0.939 |
| up_blocks.1.2.1 | 64×64 | 0.150 | 0.292 | 0.448 | 0.647 | 0.527 | 0.733 | 0.856 | 0.938 |

proportions of the top n% singular values exhibit highly consistent patterns across different layers. This consistency supports our claim that *larger $\sigma_i$ tends to correspond to foreground-dominant components, whereas smaller $\sigma_i$ is more closely associated with background information* (Sec. 2.2 of the mainbody), which confirms universally across layers. Furthermore, after inverting the singular values, the resulting distributions across layers also display similar behaviors, indicating that our style-disentangled matrices $\overline{W}_k^{im}$ effectively amplify the relatively weak background components.

## E    More discussion on the impact of $\Delta_{cs}$ in Eq. (9)

As indicated in Sec.2.4 of the mainbody, to adaptively modulate the denoising timesteps between the early and late stages, the ratio of information entropy between the customized and style references is calculated, where a larger ratio indicates that the customized reference contains more meaningful semantics than the style reference, hence more timesteps in the *early* stage for image customization and fewer timesteps in the *late* stage for style transfer are determined. To validate this, we present the generation results of joint image customization and style transfer at different timesteps. As shown in the first three rows of Fig.7, we observe that in the first and third rows, the dogs recover the details from the customized reference at the early timesteps (*e.g.*, the red clothing on the dog in the first row and the tongue of the dog in the third row), whereas in the second row, the yellow shirt on the dog is restored only at the late timesteps. In contrast, under different text prompts and style reference, the last three rows exhibit the opposite trend. Thus, by adaptively modulating the denoising timesteps between the early and late stages, our method can better generate the image according to different customized and style references.

Based on the above, we further clarify the definitions of $H_{cu}$ and $H_{st}$ in Eq.(9), where the key intuition is that **unlike the sparsity of text prompts, the differences in information entropy among images are generally modest**. Therefore, the threshold $\Delta_{cs}$ tends to stay within a relatively stable range rather than experiencing large fluctuations. To validate this intuition, we further conduct ablation studies by varying the proportion of the early denoising stage, dividing the entire denoising process into early and late stages according to the ratios {0.1, 0.2, 0.3, 0.4, 0.5}. We also report the performance of adaptively modulating the early and late stages using **Eq. (9) of the mainbody**, with detailed results shown in Table. 8.

The results indicate that when the early-stage proportion is set to 0.1, the generated images achieve the highest similarity to the style reference (CLIP-S and DINO-S), but the lowest similarity to the customized reference (CLIP-T, CLIP-C, and DINO-C). Conversely, when the proportion is increased to 0.5, the opposite trend is observed. Moreover, the performance of **DRP-Diff** falls between the results of the 0.4 and 0.5 settings, demonstrating that **our adaptive modulation strategy effectively regulates the ratio between early and late stages, achieving a balanced performance in both text alignment and dual-image similarity** (Mean(CLIP) and Mean(DINO)), which is consistent with the intuition in **Sec.2.4 of the mainbody**.

## F    More details about the implementation details

As mentioned in Sec.3.1 of the mainbody, due to page limitation, we further provide more details about the implementation details as follows:

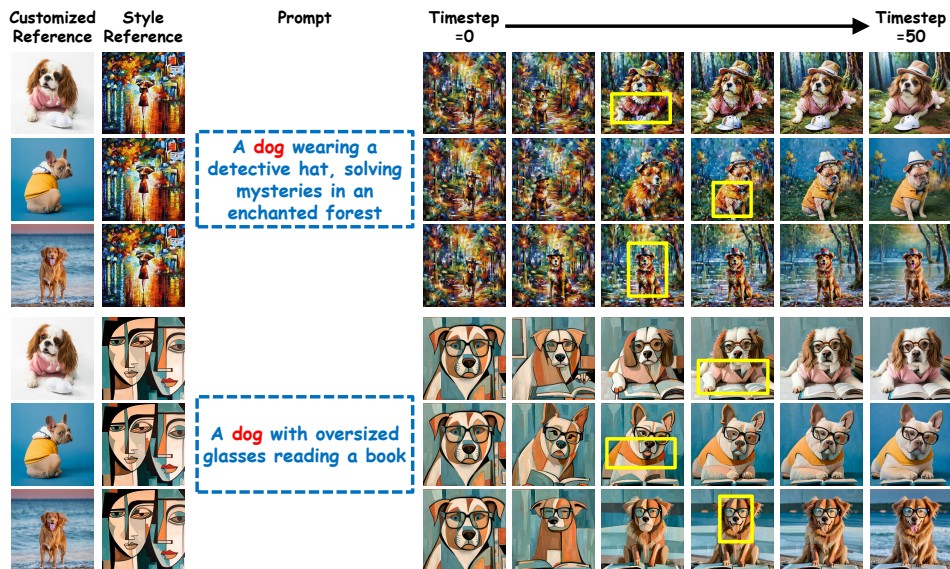

Figure 7: The personalized image of different length for the early and late stage, by adaptively modulating the denoising timesteps between the early and late stages, our DRP-Diff can better generate the image according to different customized and style references.

Table 8: Ablation studies about the influence of different denoising timesteps between the early and late stages on generating the personalized images.

| Method | CLIP-T ↑ | CLIP-C ↑ | CLIP-S ↑ | DINO-C ↑ | DINO-S ↑ | Mean(CLIP) ↑ | Mean(DINO) ↑ |
|---|---|---|---|---|---|---|---|
| 0.1 | 0.2864 | 0.6631 | 0.7036 | 0.3098 | 0.4459 | 0.5510 | 0.37785 |
| 0.2 | 0.2954 | 0.6963 | 0.6821 | 0.3830 | 0.4139 | 0.5579 | 0.39845 |
| 0.3 | 0.3035 | 0.7295 | 0.6558 | 0.4515 | 0.3682 | 0.5629 | 0.40985 |
| 0.4 | 0.3105 | 0.7554 | 0.6313 | 0.5176 | 0.3237 | 0.5657 | 0.42065 |
| 0.5 | 0.3166 | 0.7822 | 0.6104 | 0.5705 | 0.2839 | 0.5697 | 0.4262 |
| **DRP-Diff** | 0.3164 | 0.7764 | 0.6187 | 0.5609 | 0.2958 | 0.5705 | 0.42835 |

**Datasets.** We evaluate DRP-Diff on combined datasets covering image customization, style transfer, and text-to-image (T2I) generation:

- **Image Customization**: 1) **DreamBench** Ruiz et al. (2023) + **TI benchmark** Gal et al. (2022): a well-known single-object personalized image generation dataset comprising 38 subjects, covering unique objects and pets such as backpacks, stuffed animals, dogs, cats, sunglasses, and cartoons, with each subject associated with 25 prompts; 2) **DreamBench++** Peng et al. (2025): the latest human-aligned personalized image generation benchmark, comprising 65 object classes, 20 human classes, and 45 animal classes. For each image, 9 text prompts are generated by GPT-4o to span a range of difficulty levels.

- **Style Transfer**: 1) **IS-Plus benchmark** Wang et al. (2024b): a style evaluation benchmark containing 104 images across diverse styles; 2) **StyleBench** Gao et al. (2024): a style evaluation benchmark covering 73 distinct styles, ranging from paintings, flat illustrations, and 3D rendering to sculptures with varying materials. For each style, 5–7 distinct images with variations are collected. In total, StyleBench contains 490 images across diverse styles.

- **T2I generation**: **T2I-CompBench++** Huang et al. (2025a): an enhanced benchmark for compositional text-to-image generation. T2I-CompBench++ comprises 8,000 compositional text prompts categorized into four primary groups: attribute binding, object relationships, generative numeracy, and complex compositions. These are further divided into eight sub-categories.

Based on the above, we construct several combined datasets for different joint tasks: 1) for both image customization and style transfer (small-scale), we concatenate **DreamBench** and **TI benchmark** as the customization references, and use **IS-Plus benchmark** as the style references; 2) for

both style transfer and image customization (large-scale), we take **DreamBench++** as the customization references and **StyleBench** as the style references; 3) for the style transfer task, we combine **InstantStyle-Plus** and **StyleBench** as style references, while text prompts are provided by **T2I-CompBench++** for stylized image generation.

**Metrics.** We consider 9 evaluation metrics from three aspects to evaluate our DRP-Diff:

- **Text and Image Reference Alignment**: we follow previous methods and adopt two representative models, **CLIP**[3] and **DINO** Caron et al. (2021), for evaluation. Specifically, **CLIP-T** measures the similarity between generated images and the given text prompts; **CLIP-TC** and **CLIP-TS** evaluate the similarity between generated images and customized (or style) reference images while jointly considering the text prompts; finally, **CLIP-CS** and **DINO-CS** assess the similarity between generated images and dual-reference images.

- **Image Generation Quality**: we employ Image Reward (**IR**) Xu et al. (2023) to model human preferences for image quality and **HPS V2** Wu et al. (2023) that combines perceptual studies with deep learning for comprehensive visual and semantic assessment. Both metrics measure the semantics consistency of the high-quality image generation aligned with human standards;

- **Time Complexity**: We employ the inference time (**Times**) and Video Random Access Memory (**VRAM**) consumption to evaluate the computational efficiency and memory footprint of different methods during image generation.

**Model Configuration.** Our main experiments are conducted on the pre-trained IP-Adapter-Plus Ye et al. (2023) with SDXL Podell et al. (2023) model as the text-to-image diffusion model, configured with 30 sampling steps and and a fixed random seed of 42, to ensure high-quality outputs. We use the recommended hyper-parameters for each method across all images to ensure fair comparison. All experiments are implemented in PyTorch and conducted on a single NVIDIA RTX 4090 GPU.

## G ADDITIONAL QUANTITATIVE RESULTS AND QUALITATIVE ANALYSIS FOR STYLE TRANSFER

As indicated in Sec.3.2 of the mainbody, we conduct additional evaluations against state-of-the-art diffusion models for style transfer. These include **InstantStyle** Wang et al. (2024a), **CSGO** Xing et al. (2024), **StyleShot** Gao et al. (2024), **StyleAligned** Hertz et al. (2024), **DEADiff** Qi et al. (2024), **StyleCrafter** Liu et al. (2023), **MaskST** Zhu et al. (2025), **IP-Adapter** Ye et al. (2023), and **RB-Modulation** Rout et al. (2025). However, these methods often suffer from style leakage, leading to misalignment with text prompts. In addition, **StyleStudio** Lei et al. (2025) employs layout to preserve the foreground, but it still struggles to handle text prompts involving multiple objects.

**Quantitative and Qualitative analysis.** We employ the more complex text prompts in **T2I-CompBench++** to generate the style transfer result in Table.9, It is observed that **DRP-Diff** enjoys larger CLIP-TS, CLIP-S and DINO-S than the competitors in different kinds of compositional text prompts. Notably, Our DRP-Diff maintains achieves the large performance gain over **StyleAligned**, which indicates that DRP-Diff outperforms by *disentangle the foreground and background references* (Sec. 1 of the mainbody). To shed more light on the advantages of our method, we further perform the generation analysis on the style transfer results. Fig.8 delivers the following: due to the the dense style information intended for the background often leaks into the foreground, resulting in misalignment with the text prompts. **IP-Adapter** Ye et al. (2023), **CSGO** Xing et al. (2024), **Style-Crafter** Liu et al. (2023) and **MaskST** Zhu et al. (2025) inevitably generate the irrelevant semantics, *e.g.*, the extra woman from the style reference (the last rows in Fig.8). Note that, **StyleStudio** struggles to handle the text prompts with multi-objects, *e.g.*, the incorrect number of "*woman*" in the third row of Fig.8, which attributes to the *self-attention score maps mainly highlight only the most salient regions, preventing it from handling the complex text prompts* (Sec. 1 of the mainbody).

---

[3]Since CLIP is trained on real image–text (T) pairs, it is particularly sensitive to image style, which may distort text–image similarity. To alleviate this, we propose CLIP-TS by measuring similarity with style reference (S), thereby reducing the style influence . In addition, the generated background aligned with text prompts can interfere with the customized reference (C); hence we propose CLIP-TC, which incorporates text prompts to suppress such interference. We also provide CLIP-T (image–text similarity) and CLIP-CS (image–image reference similarity) for comparison.

Table 9: Quantitative comparisons for style transfer between DRP-Diff and other diffusion-based personalization models on T2I-CompBench++. ▇ and ▇ denote the best and second-best results, respectively. DRP-Diff consistently achieves the best performance.

| Category | Metric | STYLE TRANSFER METHODS | | | | | | | | |
| --- | --- | --- | --- | --- | --- | --- | --- | --- | --- | --- |
| | | InstantStyle Arxiv'24 | CSGO Arxiv'24 | StyleShot Arxiv'24 | StyleAligned CVPR'24 | DEADiff CVPR'24 | StyleCrafter TOG'24 | MaskST ICLR'25 | StyleStudio CVPR'25 | DRP-Diff |
| Attribute-Color | CLIP-TS ↑ | 0.4587 | 0.4539 | 0.4644 | 0.4832 | 0.4448 | 0.4771 | 0.4641 | 0.4497 | 0.4978 |
| | CLIP-S ↑ | 0.5967 | 0.6392 | 0.6782 | 0.6904 | 0.6177 | 0.6304 | 0.6763 | 0.5859 | 0.7188 |
| | DINO-S ↑ | 0.2146 | 0.3924 | 0.4769 | 0.4437 | 0.3318 | 0.3541 | 0.4764 | 0.2370 | 0.5007 |
| Attribute-Shape | CLIP-TS ↑ | 0.4507 | 0.4561 | 0.4595 | 0.4797 | 0.4392 | 0.4656 | 0.4597 | 0.4504 | 0.4907 |
| | CLIP-S ↑ | 0.5864 | 0.6357 | 0.6602 | 0.7051 | 0.5986 | 0.6274 | 0.6602 | 0.6162 | 0.7202 |
| | DINO-S ↑ | 0.2087 | 0.3711 | 0.4374 | 0.4658 | 0.2944 | 0.3843 | 0.4368 | 0.3087 | 0.4779 |
| Attribute-Texture | CLIP-TS ↑ | 0.4524 | 0.4561 | 0.4543 | 0.4819 | 0.4304 | 0.4712 | 0.4534 | 0.4443 | 0.4946 |
| | CLIP-S ↑ | 0.5879 | 0.6406 | 0.6553 | 0.7246 | 0.5835 | 0.6348 | 0.6538 | 0.5889 | 0.7344 |
| | DINO-S ↑ | 0.2131 | 0.3760 | 0.4476 | 0.4844 | 0.2707 | 0.3729 | 0.4465 | 0.2616 | 0.4975 |
| Relationships-Non-spatial | CLIP-TS ↑ | 0.4424 | 0.4553 | 0.4504 | 0.4839 | 0.4402 | 0.4553 | 0.4504 | 0.4314 | 0.4932 |
| | CLIP-S ↑ | 0.5723 | 0.6279 | 0.6387 | 0.7251 | 0.5972 | 0.6064 | 0.6387 | 0.5566 | 0.7207 |
| | DINO-S ↑ | 0.2336 | 0.4278 | 0.4853 | 0.5171 | 0.3533 | 0.3795 | 0.4854 | 0.2646 | 0.5274 |
| Relations-2D Spatial | CLIP-TS ↑ | 0.4568 | 0.4670 | 0.4661 | 0.4912 | 0.4563 | 0.4829 | 0.4668 | 0.4565 | 0.5063 |
| | CLIP-S ↑ | 0.5835 | 0.6309 | 0.6592 | 0.7129 | 0.6362 | 0.6421 | 0.6597 | 0.6060 | 0.7319 |
| | DINO-S ↑ | 0.2262 | 0.4113 | 0.4615 | 0.4998 | 0.3824 | 0.3977 | 0.4647 | 0.3253 | 0.5249 |
| Relationships-3D Spatial | CLIP-TS ↑ | 0.4573 | 0.4624 | 0.4663 | 0.4875 | 0.4497 | 0.4773 | 0.4663 | 0.4429 | 0.5029 |
| | CLIP-S ↑ | 0.5918 | 0.6333 | 0.6680 | 0.7100 | 0.6333 | 0.6357 | 0.6680 | 0.5732 | 0.7334 |
| | DINO-S ↑ | 0.2166 | 0.3819 | 0.4654 | 0.4760 | 0.3655 | 0.3774 | 0.4696 | 0.2249 | 0.5099 |
| Numeracy | CLIP-TS ↑ | 0.4490 | 0.4509 | 0.4597 | 0.4902 | 0.4465 | 0.4783 | 0.4597 | 0.4436 | 0.4946 |
| | CLIP-S ↑ | 0.5845 | 0.6182 | 0.6562 | 0.7241 | 0.6226 | 0.6499 | 0.6558 | 0.6011 | 0.7212 |
| | DINO-S ↑ | 0.2289 | 0.3882 | 0.4779 | 0.5116 | 0.3541 | 0.4346 | 0.4738 | 0.3160 | 0.5182 |
| Complex | CLIP-TS ↑ | 0.4404 | 0.4500 | 0.4556 | 0.4836 | 0.4390 | 0.4636 | 0.4558 | 0.4351 | 0.4937 |
| | CLIP-S ↑ | 0.5864 | 0.6445 | 0.6582 | 0.7256 | 0.6021 | 0.6387 | 0.6582 | 0.5791 | 0.7319 |
| | DINO-S ↑ | 0.2134 | 0.3911 | 0.4533 | 0.4961 | 0.2938 | 0.3956 | 0.4536 | 0.2274 | 0.5027 |

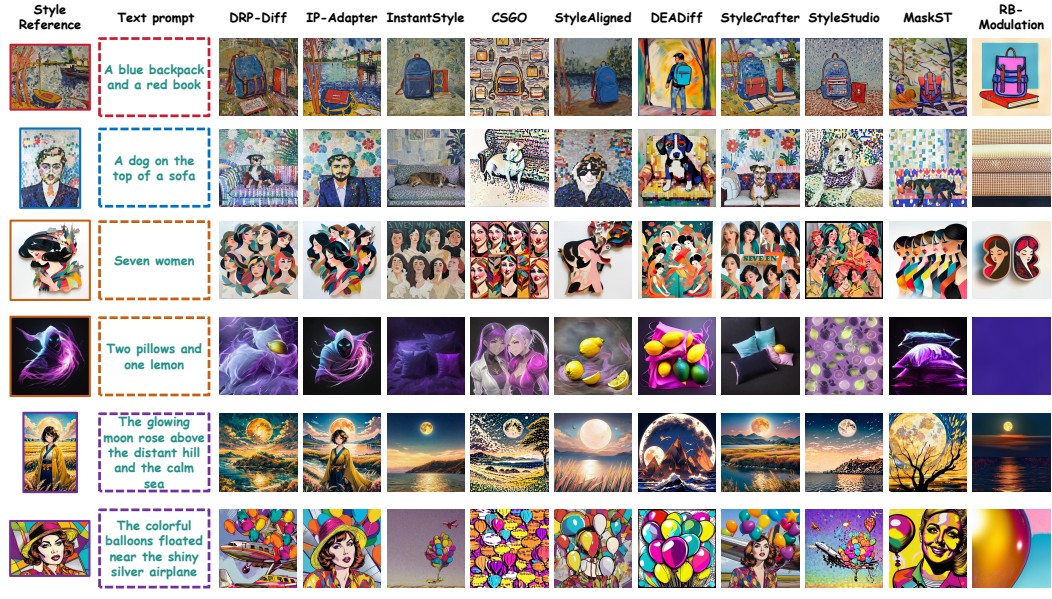

Figure 8: Comparison of style transfer with state-of-the-art methods. DRP-Diff delivers superior personalized results over competing approaches.

# H  ADDITIONAL GENERATED RESULTS ABOUT COMPLEXNESS FOR INTER-OBJECT RELATIONSHIP FOR STYLE TRANSFER

We further provide the quantitative relationship between object count and performance metrics, along with the detailed explanations, as shown in Table 10. We conduct additional ablation studies by reducing the number of objects described in the text prompts from the numeracy category of

**T2I-CompBench++**, thereby forming prompts of different difficulty levels: **Easy (0–3 objects)**, **Medium (4–6 objects)**, and **Hard (more than 6 objects)**.

The quantitative results show that as the text prompts increase in complexity—from Easy to Medium to Hard—the generation task becomes inherently more challenging. **Text prompts containing more objects impose stronger compositional constraints on the model, making it progressively harder to maintain consistency with both the text prompts and the style reference.** Consequently, performance across all metrics shows a slight decline. This observation indicates that **DRP-Diff maintains strong performance by effectively disentangling the foreground and background references** (Sec. 1 of the mainbody).

Table 10: Quantitative analysis across different difficulty levels determined by the number of objects in text prompts.

| Difficulty | CLIP-TS ↑ | CLIP-S ↑ | DINO-S ↑ |
|---|---|---|---|
| Easy | 0.4966 | 0.7310 | 0.5313 |
| Medium | 0.4954 | 0.7261 | 0.5273 |
| Hard | 0.4951 | 0.7236 | 0.5263 |

Based on the above, we further evaluate the generative numeracy performance of our **DRP-Diff** compared with **CountGen** Binyamin et al. (2025). For a fair comparison, we use natural scenes as the style reference in **DRP-Diff** and vary the number of objects specified in the text prompts from the numeracy category of T2I-CompBench++, resulting in three difficulty levels: Easy (0–3 objects), Medium (4–6 objects), and Hard (more than 6 objects). We employ **CLIP-T**, Image Reward (IR), HPS, and Aesthetic Score (AS) to assess, respectively, the alignment between generated images and the text prompts, the overall quality of the generated images, and their aesthetic appeal. The quantitative results of Table.11, summarized below, show that as the text prompts increase in complexity—from Easy to Medium to Hard—our method outperforms CountGen. This validates that **our approach can effectively address the issue of dense background-oriented style information leaking into the foreground, resulting in misalignment with the text prompts** (Sec. 1 of the mainbody). We also find that **CountGen is limited to generating multiple objects of the same category**; it fails when the text contains multiple objects of different categories. Moreover, **CountGen requires 36s on an A100 GPU for inference, whereas our method only needs 5.66s on an RTX 4090 GPU.**

Table 11: Comparison of CountGen and DRP-Diff across different difficulty levels of generative numeracy.

| Method | CLIP-T↑ | IR$_{\times 10}$↑ | HPS$_{\times 10^2}$↑ | AS↑ |
|---|---|---|---|---|
| Easy-CountGen | 0.3008 | 2.3655 | 25.8822 | 3.0522 |
| Easy-**DRP-Diff** | 0.3049 | 5.6709 | 27.7889 | 3.1894 |
| Medium-CountGen | 0.2981 | 2.4647 | 25.6348 | 3.1733 |
| Medium-**DRP-Diff** | 0.2998 | 3.8341 | 27.4577 | 3.2562 |
| Hard-CountGen | 0.2878 | -2.2082 | 24.0633 | 3.2227 |
| Hard-**DRP-Diff** | 0.2971 | 4.8485 | 26.7440 | 3.2925 |

## I  ADDITIONAL QUANTITATIVE RESULTS AND QUALITATIVE ANALYSIS FOR JOINT IMAGE CUSTOMIZATION AND STYLE TRANSFER

As indicated in Sec.3.2 of the mainbody, we further exhibit additional quantitative results by performing the experiments applying **CSGO** Xing et al. (2024) and **StyleID** Chung et al. (2024), to transfer the styles of various image customization methods, including **IP-Adapter** Ye et al. (2023), **Dreamcache** Aiello et al. (2025), **PatchDPO** Huang et al. (2025b), **SSR-Encoder** Zhang et al. (2024), **MIP-Adapter** Huang et al. (2025b), and **MS-Diffusion** Wang et al. (2025), as shown in Table 12, [**MS-Diffusion** Wang et al. (2025) + **StyleID** Chung et al. (2024)] remains the large performance margins at most 2.41% for CLIP-T, 8.26% for CLIP-TC 0.82% for CLIP-TC, 6.4% for

Table 12: Quantitative comparisons for both image customization and style transfer between DRP-Diff and other diffusion-based personalization models on DreamBooth, TI benchmark, and Dream-Booth++. ▇ and ▇ denote the best and second-best results, respectively. DRP-Diff consistently achieves the best performance.

| | DreamBooth and TI benchmark | | | | | | | DreamBooth++ | | | | | | |
|---|---|---|---|---|---|---|---|---|---|---|---|---|---|---|
| | IP-Adapter Arxiv'23 | SSR-Encoder CVPR'24 | MIP-Adapter AAAI'25 | Dreamcache CVPR'25 | PatchDPO CVPR'25 | MS-Diffusion ICLR'25 | DRP-Diff | IP-Adapter Arxiv'23 | SSR-Encoder CVPR'24 | MIP-Adapter AAAI'25 | Dreamcache CVPR'25 | PatchDPO CVPR'25 | MS-Diffusion ICLR'25 | DRP-Diff |
| | {BASELINE} + CSGO (ARXIV'24) | | | | | | | {BASELINE} + CSGO (ARXIV'24) | | | | | | |
| CLIP-T ↑ | 0.2854 | 0.2822 | 0.2810 | 0.2705 | 0.2778 | 0.3120 | 0.3196 | 0.2849 | 0.2837 | 0.2808 | 0.2900 | 0.2808 | 0.3125 | 0.3108 |
| CLIP-TC ↑ | 0.4670 | 0.4561 | 0.4612 | 0.4360 | 0.4602 | 0.4548 | 0.4875 | 0.4934 | 0.4817 | 0.4875 | 0.4675 | 0.4873 | 0.4812 | 0.5122 |
| CLIP-TS ↑ | 0.4434 | 0.4448 | 0.4448 | 0.4517 | 0.4485 | 0.4612 | 0.4612 | 0.4395 | 0.4431 | 0.4392 | 0.4534 | 0.4395 | 0.4541 | 0.4558 |
| CLIP-CS ↑ | 0.6606 | 0.6543 | 0.6572 | 0.6431 | 0.6597 | 0.6401 | 0.6714 | 0.6709 | 0.6631 | 0.6680 | 0.6499 | 0.6680 | 0.6436 | 0.6821 |
| DINO-CS ↑ | 0.3873 | 0.3832 | 0.3875 | 0.3604 | 0.3857 | 0.3628 | 0.4089 | 0.4132 | 0.3889 | 0.4149 | 0.3736 | 0.4053 | 0.3636 | 0.4262 |
| IR×10 ↑ | -4.8666 | -7.2286 | -5.7244 | -8.1185 | -6.7291 | 4.7799 | 6.9093 | -6.2800 | -7.6196 | -7.3266 | -5.4379 | -7.5620 | 3.6248 | 1.9510 |
| HPS×10 ↑ | 2.3648 | 2.2943 | 2.2715 | 2.1386 | 2.2601 | 2.5678 | 2.5960 | 2.3154 | 2.2441 | 2.2533 | 2.2201 | 2.2385 | 2.5194 | 2.5260 |
| Time(s) ↓ | 17.39 | 15.12 | 19.94 | 16.26 | 17.20 | 18.77 | 5.66[10.54] | 17.39 | 15.12 | 19.94 | 16.26 | 17.20 | 18.77 | 5.66[10.54] |
| VRAM(GB) ↓ | 4.72 | 1.68 | 4.62 | 4.72 | 4.72 | 4.65 | 4.55 | 4.72 | 1.68 | 4.62 | 4.72 | 4.72 | 4.65 | 4.55 |
| | {BASELINE} + STYLEID (CVPR'24) | | | | | | Ours | {BASELINE} + STYLEID (CVPR'24) | | | | | | Ours |
| CLIP-T ↑ | 0.2710 | 0.2744 | 0.2988 | 0.2505 | 0.2612 | 0.2988 | 0.3196 | 0.2751 | 0.2793 | 0.2708 | 0.2788 | 0.2688 | 0.3035 | 0.3108 |
| CLIP-TC ↑ | 0.4724 | 0.4690 | 0.4497 | 0.4275 | 0.4631 | 0.4497 | 0.4875 | 0.4932 | 0.4856 | 0.4900 | 0.4573 | 0.4858 | 0.4731 | 0.5122 |
| CLIP-TS ↑ | 0.4358 | 0.4390 | 0.4590 | 0.4485 | 0.4377 | 0.4590 | 0.4612 | 0.4341 | 0.4351 | 0.4326 | 0.4492 | 0.4363 | 0.4521 | 0.4558 |
| CLIP-CS ↑ | 0.6714 | 0.6694 | 0.6436 | 0.6470 | 0.6704 | 0.6436 | 0.6714 | 0.6748 | 0.6636 | 0.6733 | 0.6450 | 0.6748 | 0.6411 | 0.6821 |
| DINO-CS ↑ | 0.4272 | 0.4041 | 0.3783 | 0.3667 | 0.4176 | 0.3783 | 0.4089 | 0.4389 | 0.4005 | 0.4463 | 0.3766 | 0.4357 | 0.3714 | 0.4262 |
| IR×10 ↑ | -8.9082 | -9.4933 | -10.0956 | -12.0005 | -11.1300 | 3.6003 | 6.9093 | -8.2948 | -8.0270 | -9.0297 | -7.8884 | -9.8165 | 1.6967 | 1.9510 |
| HPS×10 ↑ | 2.0695 | 2.0708 | 1.9670 | 1.8305 | 1.9763 | 2.3233 | 2.5960 | 2.1028 | 2.1251 | 2.0476 | 1.9947 | 2.0194 | 2.3410 | 2.5260 |
| Time(s) ↓ | 13.30 | 11.03 | 15.85 | 12.17 | 13.11 | 14.68 | 5.66[10.54] | 13.30 | 11.03 | 15.85 | 12.17 | 13.11 | 14.68 | 5.66[10.54] |
| VRAM(GB) ↓ | 16.92 | 13.88 | 16.82 | 13.80 | 16.92 | 16.85 | 4.55 | 16.92 | 13.88 | 16.82 | 13.80 | 16.92 | 16.85 | 4.55 |
| | {BASELINE} + STYLESSP (CVPR'25) | | | | | | Ours | {BASELINE} + STYLESSP (CVPR'25) | | | | | | Ours |
| CLIP-T ↑ | 0.2651 | 0.2664 | 0.2590 | 0.2490 | 0.2588 | 0.2937 | 0.3196 | 0.2632 | 0.2649 | 0.2578 | 0.2632 | 0.2595 | 0.2876 | 0.3108 |
| CLIP-TC ↑ | 0.4543 | 0.4783 | 0.4504 | 0.4241 | 0.4531 | 0.4451 | 0.4875 | 0.4785 | 0.4675 | 0.4727 | 0.4468 | 0.4744 | 0.4629 | 0.5122 |
| CLIP-TS ↑ | 0.4426 | 0.4443 | 0.4470 | 0.4482 | 0.4451 | 0.4631 | 0.4612 | 0.4368 | 0.4370 | 0.4348 | 0.4453 | 0.4375 | 0.4534 | 0.4558 |
| CLIP-CS ↑ | 0.6636 | 0.6592 | 0.6670 | 0.6426 | 0.6685 | 0.6460 | 0.6714 | 0.6724 | 0.6592 | 0.6685 | 0.6440 | 0.6714 | 0.6465 | 0.6821 |
| DINO-CS ↑ | 0.4088 | 0.3864 | 0.4036 | 0.3588 | 0.3988 | 0.3709 | 0.4089 | 0.4267 | 0.3901 | 0.4213 | 0.3721 | 0.4204 | 0.3734 | 0.4262 |
| IR×10 ↑ | -10.5572 | -10.4534 | -12.1468 | -11.7708 | -12.5709 | 1.4488 | 6.9093 | -11.1308 | -10.4540 | -12.0027 | -10.4016 | -11.6970 | -0.4119 | 1.9510 |
| HPS×10 ↑ | 2.0392 | 2.0346 | 1.9178 | 1.8243 | 1.9353 | 2.2801 | 2.5960 | 1.9782 | 1.9959 | 1.8972 | 1.8881 | 1.9054 | 2.1933 | 2.5260 |
| Time(s) ↓ | 9.61 | 7.34 | 12.16 | 8.48 | 9.42 | 10.99 | 5.66[10.54] | 9.61 | 7.34 | 12.16 | 8.48 | 9.42 | 10.99 | 5.66[10.54] |
| VRAM(GB) ↓ | 4.63 | 1.59 | 4.53 | 1.51 | 4.63 | 4.56 | 4.55 | 4.63 | 1.59 | 4.53 | 1.51 | 4.63 | 4.56 | 4.55 |

CLIP-CS 14.75% for DINO-CS 14.99% for IR and 7.9% for HPS) compared to **DRP-Diff** in Table. 12, the results verifies the intuition in Sec. 1 of the mainbody – *the dense image references that attend only to either the foreground or background, while overwhelming the sparse text prompts.*

We further report the additional inference time and memory usage introduced by our method, compared to the **IP-Adapter** that uses a single image reference, our **DRP-Diff** with two image references incurs only an additional 0.91s in inference time and 0.55GB in peak VRAM usage in Table. 13. This is because **Eq. (4)** and **Eq. (5)** of the main body are parameter-dependent and independent of the input data, allowing them to be pre-computed during the model loading stage. As a result, this computation cost is eliminated during inference, which is consistent with the design described in **Sec. 2.2** of the main body.

Table 13: The additional inference time and VRAM consumption comparison.

| Method | Time(s)↓ | VRAM(GB)↓ |
|---|---|---|
| **IP-Adapter** | **4.75** | **4.00** |
| **DRP-Diff** | 5.66 | 4.55 |

To shed more light on the advantages of our method, we further perform the generation analysis on the personalized generated results with *the higher resolution*. Fig.9 delivers the following: due to the entanglement of foreground and background, **RB-Modulation** Rout et al. (2025) struggles to generate customized content (see the first row in Fig.9). In addition, **IP-Adapter** Ye et al. (2023), **Dreamcache** Aiello et al. (2025), and **PatchDPO** Huang et al. (2025b) often exhibit unreasonable results that fail to align with the text prompts, *e.g.*, the missing guitar in the 4th row of IP-Adapter and the missing butterfly in the first row of Dreamcache and PatchDPO in Fig.9, which attributes to the *the dense image references that attend only to either the foreground or background, while overwhelming the sparse text prompts.* (Sec.1 of the mainbody).

## J    MORE GENERATION RESULTS OF DIFFERENT T2I MODELS

As mentioned in Sec. 3.2 of the mainbody, we further report the generated results of several representative text-to-image (T2I) models, including different finetuned variants of Stable Diffusion

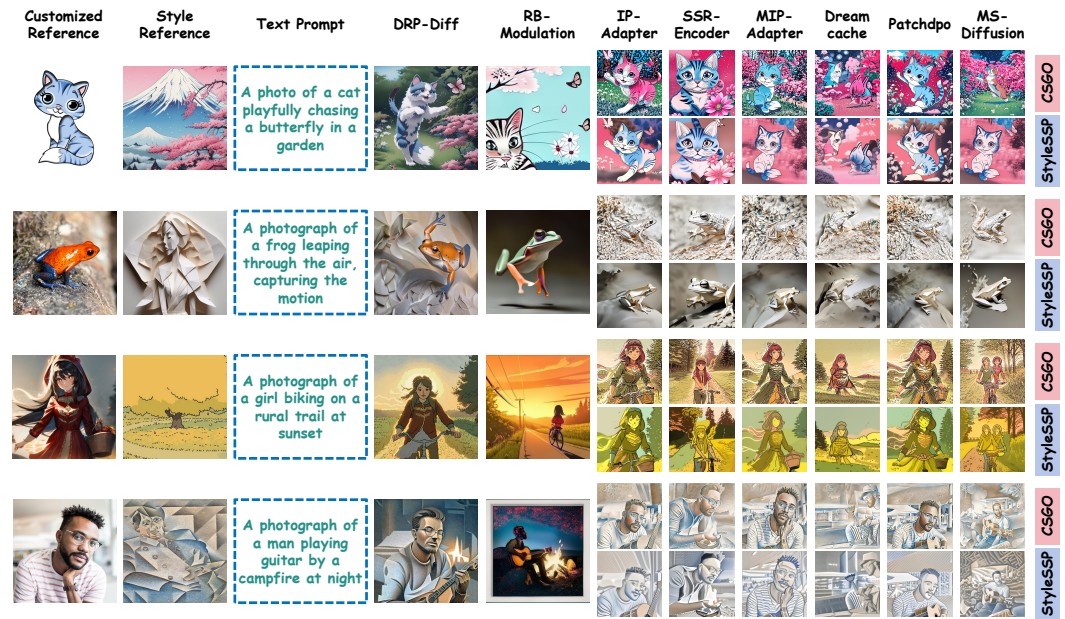

Figure 9: Comparison of joint image customization and style transfer with state-of-the-art methods. DRP-Diff delivers superior personalized results over competing approaches.

XL (SDXL), namely Realistic VisionXL[4] and DreamshaperXL[5], as well as the DiT-based approach HunyuanDiT[6]

These three models exhibit distinct characteristics due to differences in their training datasets. Specifically,

- **Realistic VisionXL**: fine-tuned on high-quality, photography-oriented datasets, emphasizing realism, lighting, and fine-grained textures, making it particularly effective for photorealistic inpainting tasks. As shown in Fig. 10(a), the customized contents and the style can achieve a better balance, which makes it suitable for either image customization or style transfer.
- **DreamshaperXL**: incorporates a mixture of artistic and stylized datasets, leading to visually rich outputs with enhanced creativity and aesthetic variations, though sometimes at the expense of strict semantic fidelity to the input prompts. As shown in Fig. 10(b), the personalized images emphasize style more strongly, making it more competent for the style transfer of our DRP-Diff.
- **HunyuanDiT**: trained on large-scale, multi-domain datasets covering both natural scenes and human-centric images, which provides stronger generalization ability across diverse prompts and better semantic alignment. Benefiting from its realism-oriented training, it is more favorable to the image customization of our DRP-Diff, as illustrated in Fig. 10(c).

## K CODE

We implement the proposed DRP-Diff in pytorch framework under the running environment as: python3.6, pytorch1.7 and cuda10.0. The codes are available in the Supplementary Material with the folder name **DRP-Diff**.

## L ETHICS STATEMENT

This work adheres to the ICLR Code of Ethics. In this study, no human subjects or animal experimentation was involved. All datasets used, including **DreamBench** Ruiz et al. (2023), **TI**

---

[4] https://civitai.com/models/4201/realistic-vision-v60-b1
[5] https://civitai.com/models/112902/dreamshaper-xl
[6] https://huggingface.co/Tencent-Hunyuan/IP-Adapter

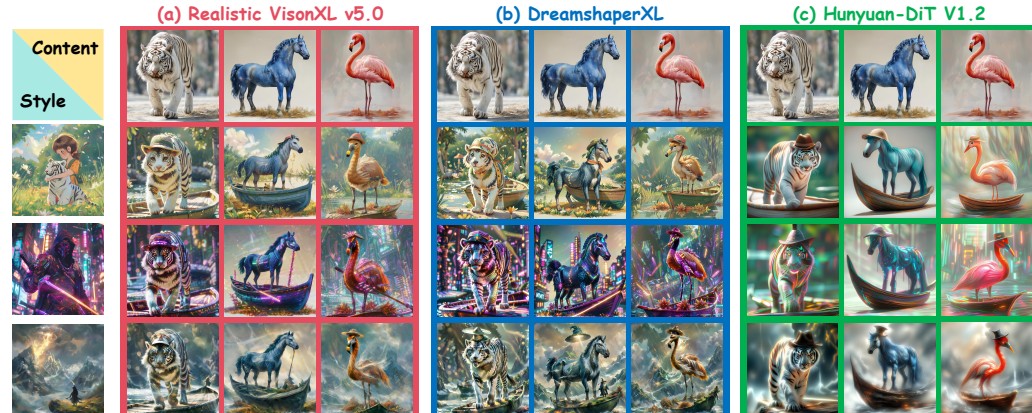

Figure 10: Generation results of different T2I models.

**benchmark** Gal et al. (2022), **DreamBench++** Peng et al. (2025), **IS-Plus** Wang et al. (2024b), **StyleBench** Gao et al. (2024) and **T2I-CompBench++** Huang et al. (2025a) were sourced in compliance with relevant usage guidelines, ensuring no violation of privacy. We have taken care to avoid any biases or discriminatory outcomes in our research process. No personally identifiable information was used, and no experiments were conducted that could raise privacy or security concerns. We are committed to maintaining transparency and integrity throughout the research process.

## M  REPRODUCIBILITY STATEMENT

We have made every effort to ensure that the results presented in this paper are reproducible. All code and datasets have been made publicly available in an anonymous repository to facilitate replication and verification. The experimental setup, including training steps, model configurations, and hardware details, is described in detail in the paper. We have also provided a full description of our **DRP-Diff**, to assist others in reproducing our experiments. Additionally, all datasets (**DreamBench** Ruiz et al. (2023), **TI benchmark** Gal et al. (2022), **DreamBench++** Peng et al. (2025), **IS-Plus** Wang et al. (2024b), **StyleBench** Gao et al. (2024) and **T2I-CompBench++** Huang et al. (2025a)) are publicly available, ensuring consistent and reproducible evaluation results. We believe these measures will enable other researchers to reproduce our work and further advance the field.

## N  THE USE OF LARGE LANGUAGE MODELS

Large Language Models (LLMs) were used to aid in the writing and polishing of the manuscript. Specifically, we used an LLM to assist in refining the language, improving readability, and ensuring clarity in various sections of the paper. The model helped with tasks such as sentence rephrasing, grammar checking, and enhancing the overall flow of the text.

It is important to note that the LLM was not involved in the ideation, research methodology, or experimental design. All research concepts, ideas, and analyses were developed and conducted by the authors. The contributions of the LLM were solely focused on improving the linguistic quality of the paper, with no involvement in the scientific content or data analysis.

The authors take full responsibility for the content of the manuscript, including any text generated or polished by the LLM. We have ensured that the LLM-generated text adheres to ethical guidelines and does not contribute to plagiarism or scientific misconduct.

