# OpenReview forum: "Taming Text Alignment for Personalization: Disentangling Foreground Customization and Background Style in Diffusion Models for Personalized Image Generation"
_ICLR.cc/2026/Conference — Submitted to ICLR 2026_

### Official Review · Reviewer_rfeM · 2025-10-23

**Soundness:** 2
**Presentation:** 1
**Contribution:** 1
**Rating:** 2
**Confidence:** 3

**Summary:**

The authors proposed a customization method of a diffusion model, focusing on objects and style. The method requires a text prompt, object reference images, and style reference images separately. They also propose to disentangle the foreground and background of images and process them separately. The suggestion is to have different procedures for different sides of the image.

**Strengths:**

1. The only strength I see in this paper is the approach (not the idea) of disentanglement of foreground and background with the SVD method.

**Weaknesses:**

1. Poor problem formulation - not justified separation of diffusion model customization and style transfer. In both cases, the model gets customized to concepts, and the concepts are either objects or style.

2. The idea of background and foreground disentanglement is artificial and unnatural. No justification for why this method is superior. No justification for why to formulate image customization for foreground and style transfer for background.

3. The problem mentioned in 083 - incorrect number of “woman” - can have different roots. That is not proof of the problem the authors described.  e.g., in the paper Make It Count [1], the diffusion model struggles to keep the count correct in generated images. They do not deal with any style transfer.

4. Unclear logical chain in 086-089. It needs more evidence to make it true.

5. The proposed method requires 2 types of reference images (dual reference): object images and style images.

6. The method is complex.

7. Poor fidelity of model customization on concepts in Figure 5. Specifically, the robot's attributes are very different in the original and DRP-Diff generated images: arm color, legs, and texture on the belly - all are different. Same thing with the butterfly example - there is no red texture on the original one.

8. One of the most significant weaknesses is a very bad, unclear, and complex writing style: even the known knowledge is written badly with complex formulations. (e.g., Figure 3 caption, Table 1 b caption, method section, etc.). Also, poor visualization for tables and figures in quantitative and qualitative comparisons. They are very small, badly organized. Its customization, the visual attributes should be investigated properly.

[1] Make It Count: Text-to-Image Generation with an Accurate Number of Objects
Lital Binyamin, Yoad Tewel, Hilit Segev, Eran Hirsch, Royi Rassin, Gal Chechik
The IEEE/CVF Conference on Computer Vision and Pattern Recognition (CVPR)

**Questions:**

1. What are “sparse text prompt” and “dense text prompt”?
2. How background of F get disentangled in the noise latent code (Fig. 2 caption)?

---

> ### Author Response · Authors · 2025-11-22
> **Author Response (part 1 of 3)**
>
> # To Reviewer rfeM
> Thanks very much for your valuable comments that helped us significantly improve this submission. **We kindly remind that, as per the rebuttal policy, we also submitted our refined manuscript**. Below are detailed point-to-point responses about how we address your questions.
>
> ---
> # [W1]: Question on the Relation Between Image Customization and Style Transfer
>
> **Response**: Thanks for the valuable comments. We would like to clarify that, as mentioned in **Lines 99-103, Sec.1 of the mainbody**: *both image customization and style transfer exhibit suboptimal alignment with the text prompts. In particular, image customization primarily suffers from background misalignment, whereas style transfer is prone to foreground misalignment. __This misalignment arises from the dense image references that attend only to either the foreground or background, while overwhelming the sparse text prompts.__*, thus motivating us to propose a dual-reference personalization diffusion model, named DRP-Diff, for both image customization and style transfer to achieve the text alignment for personalized image generation.
>
> To validate the above intuition, We provide quantitative comparisons to demonstrate that our **DRP-Diff** is effective for both image customization and style transfer. As indicated in **Table.1(b) of the mainbody, Lines 1007-1020, Sec.G of the Appendix and Table.9 of the Appendix, we copy that as follows**:
>
> * **Table.1(b) of the mainbody** for image customization
>
> |Metric|TI|DreamBooth|Custom-Diff|IP-Adapter|SSR-Encoder|BLIP-Diff|MS-Diff|Subject-Diff|JeDi|DRP-Diff|
> |:---:|:---:|:---:|:---:|:---:|:---:|:---:|:---:|:---:|:---:|:---:|
> |**DINO**|0.569|0.668|0.643|0.608|0.612|0.594|0.671|**0.711**|0.679|0.707|
> |**CLIP-I**|0.780|0.803|0.790|0.809|0.821|0.779|0.792|0.787|0.814|**0.833**|
> |**CLIP-T**|0.255|0.305|0.305|0.274|0.308|0.300|0.321|0.293|0.293|**0.323**|
> |**Avg.**|0.535|0.592|0.579|0.564|0.580|0.558|0.595|0.597|0.595|**0.621**|
> |||
>
> * **Table.9 of the Appendix (Partial results)** for style transfer
>
> |Metric|InstantStyle|CSGO|StyleShot|StyleAligned|DEADiff|StyleCrafter|MaskST|StyleStudio|DRP-Diff|
> |:---:|:---:|:---:|:---:|:---:|:---:|:---:|:---:|:---:|:---:|
> |**CLIP-TS ↑**|0.4587|0.4539|0.4644|0.4832|0.4448|0.4771|0.4641|0.4497|**0.4978**|
> |**CLIP-S ↑**|0.5967|0.6392|0.6782|0.6904|0.6177|0.6304|0.6763|0.5859|**0.7188**|
> |**DINO-S ↑**|0.2146|0.3924|0.4769|0.4437|0.3318|0.3541|0.4764|0.2370|**0.5007**|
> |||
> |**CLIP-TS ↑**|0.4507|0.4561|0.4595|0.4797|0.4392|0.4656|0.4597|0.4504|**0.4907**|
> |**CLIP-S ↑**|0.5864|0.6357|0.6602|0.7051|0.5986|0.6274|0.6602|0.6162|**0.7202**|
> |**DINO-S ↑**|0.2087|0.3711|0.4374|0.4658|0.2944|0.3843|0.4368|0.3087|**0.4779**|
> |||
>
>  * The corresponding results analysis are clearly claimed **Lines 1007-1020, Sec.G of the Appendix** below:
>
> *"We employ the more complex text prompts in T2I-CompBench++ to generate the style transfer result in Table.9, It is observed that DRP-Diff enjoys larger CLIP-TS, CLIP-S and DINO-S  than the competitors in different kinds of  compositional text prompts. Notably, Our DRP-Diff maintains achieves the large performance gain over StyleAligned, which indicates that **DRP-Diff outperforms by disentangle the foreground and background references** (**Sec.1 of the mainbody**)"*
>
> ---
> # [W2]: Question on Foreground for Image Customization and Background for Style Transfer
>
> **Response**: Thanks for the question. We mildly remind you that we do present the discussion on why the foreground for image customization and background for style transfer. As clearly indicated in the **Lines 78-98, Sec.1 of the mainbody and Fig.1(a)(b) of the mainbody, we copy that as follows:  (please also kindly refer to **Fig.1 in the mainbody**)**:
>
> * **Lines 78-98, Sec.1 of the mainbody**
>
> *"For image customization, prior methods tend to **emphasize dense image references of the foreground while overwhelming sparse text prompts of the background**,..., Building on this, layout-based methods,..., **explicit separation of foreground and background in the layout disrupts overall consistency**, leading to misalignment with complex or long text prompts.*
>
> *From the perspective of style transfer, during the denoising process, especially in the early stage with high-level noise, the foreground is reconstructed from sparse text prompts, and **the dense style information intended for the background often leaks into the foreground**, resulting in misalignment with the text prompts..."*
>
> ---

---

> ### Author Response · Authors · 2025-11-22
> **Author Response (part 2 of 3)**
>
> # [W3]: Question on the Object Count in Style Transfer
>
> **Response**: Thanks for the valuable comments. We mildly remind you that, as mentioned in **Lines 965-969, Sec. F of the Appendix**, we utilize **T2I-CompBench++** to provide the text prompts for stylized image generation. T2I-CompBench++ comprises **8,000** compositional text prompts categorized into four primary groups: attribute binding, object relationships, generative numeracy, and complex compositions. Notably, **generative numeracy is just one subset of T2I-CompBench++; our DRP-Diff is not limited to this category, while can further generate images across all types**, as illustrated in **Sec.G and Table.9 of the Appendix**.
>
> Following your suggestions, we further conduct the experiments on exploiting the performance of generative numeracy between our **DRP-Diff** and **CountGen** [1]. To make a  fair comparison, we utilize the natural scene as the style reference in our **DRP-Diff**, along with the detailed explanations, as shown below:
>
>
> | Method            | CLIP-T↑ | IR$_{×10}$↑  | HPS$_{×10^{2}}$ ↑ |AS ↑ |
> |------------------|:----------|:---------|:----------|:----------|
> | Easy-CountGen    | 0.3008  |  2.3655 | 25.8822 | 3.0522 |
> | Easy-**DRP-Diff**| **0.3049**  |  **5.6709** | **27.7889** | **3.1894** |
> | Medium-CountGen  | 0.2981  |  2.4647 | 25.6348 | 3.1733 |
> | Medium-**DRP-Diff**| **0.2998**  |  **3.8341** | **27.4577** | **3.2562** |
> | Hard-CountGen    | 0.2878  | -2.2082 | 24.0633 | 3.2227 |
> | Hard-**DRP-Diff**| **0.2971**  |  **4.8485** | **26.7440** | **3.2925** |
> |||
>
>
> - **Analysis of the performance of generative numeracy**
>
> "*We conduct the experiment by reducing the number of objects described in the text prompts from the numeracy category of T2I-CompBench++, thereby forming prompts of different difficulty levels: __Easy (0–3 objects), Medium (4–6 objects), and Hard (more than 6 objects)__. We adopt __CLIP-T, Image Reward (IR), HPS, and Aesthetic Score (AS)__ to evaluate __the alignment between the generated images and the text prompts, the quality of the generated images, and their aesthetic scores__.*
>
> *The quantitative results, summarized below, show that as the text prompts increase in complexity—from Easy to Medium to Hard—our method outperforms CountGen. This validates that our approach can effectively address the issue of dense background-oriented style information leaking into the foreground, which is consistent with **our intuition of disentangling the foreground and background via singular value decomposition**, as identified in **Lines 226–228, Sec. 2.2 of the mainbody**. We also find that **CountGen is limited to generating multiple objects of the same category**; it fails when the text contains multiple objects of different categories. Moreover, **CountGen requires 36s on an A100 GPU for inference, whereas our method only needs 5.66s on an RTX 4090 GPU.***"
>
> **_Following your comments above, the above results as well as analysis are also made up in Sec. H of the Appendix for our refined manuscript._**
>
>
> ---
> # [W4]: More Examples on the StyleStdio
>
> **Response**: Thanks for the constructive comments. We mildly remind you that we do present lots of example to validate the conclusion in **Lines 95-98, Sec.1 of the mainbody**, i.e., due to the StyleStdio utilizes the self-attention score maps highlight the most salient regions, thus fails to align the generated image with the text prompts of the multi-objects.  As indicated in the **Fig.1 (b) of the mainbody and Fig.8 of the Appendix, we copy that as follows:**
>
> **please kindly refer to the corresponding figures in our refined manuscript**
>
> - For the first column of **Fig.1(b)** and the fourth row of **Fig.8**, StyleStdio misses "pillows" and mismatches "lemon".
> - For the second column of **Fig.1(b)**, StyleStdio mismatches "couple".
> - For the third column of **Fig.1(b)** and the third row of **Fig.8**, StyleStdio mismatches the number of "woman".
>
> ---
> # [W5]: The Clarification of the Dual Image Reference
>
> **Response**: Thanks for the valuable comments. We kindly clarify that both the customized and style references are necessary. As indicated in the **Line 99-107, Sec.1 of the mainbody, we copy that as follows:**
>
> "*So far, both image customization and style transfer exhibit suboptimal alignment with the text prompts. In particular, image customization primarily suffers from background misalignment, whereas style transfer is prone to foreground misalignment. **This misalignment arises from the dense image references that attend only to either the foreground or background, while overwhelming the sparse text prompts**.*
>
> *To this end, we propose a dual-reference personalization diffusion model, named DRP-Diff (see Fig.2(a)). Given text prompt, unlike the existing personalized image generation with single reference image as input, **DPR-Diff simultaneously handle both customized and style reference images** to achieve the goal.*"
>
> ---

---

> ### Author Response · Authors · 2025-11-22
> **Author Response (part 3 of 3)**
>
> # [W6]: The Clarification on the Methodological Complexity
>
> **Response**: Thanks. To facilitate a clearer understanding of DRP-Diff, we provide a summarized overview of the joint image customization and style transfer process, and present the full procedural details in **Algorithm 1 of the Appendix**, **we copy that as follows:**
>
> >---
> >**Algorithm: DRP-Diff**
> >---
> >---
> >**Input:**
> >- Random Gaussian noise $z_T$
> >- Text prompts $C_{te}$ (foreground and background)
> >- Text prompts of background $\overline{C}_{te}$
> >- Customized reference $C_{cu}$
> >- Style reference $C_{st}$
> >- Denoising network $\epsilon_{\theta}$
> >- Timestep $T$
> >- Decoder $\mathcal{D}$
> >---
> >**Output:**
> >- Personalized generated image $I_0$
> >---
> >1. Calculate threshold $\Delta_{cs}$ via **Eq. (9) of the mainbody**
> >2. **For** $t = T$ **to** $\frac{1}{1+\Delta_{cs}}T$ **do**:
> >3.    - Compute $z_{t-1}$ from $z_t$ using decoupled cross-attention (**Eqs. (1) and (7) of the mainbody**) of $\epsilon_{\theta}$, conditioned on $C_{te}$, $C_{cu}$, $C_{st}$ and $\overline{C}_{te}$ following **Eq. (6) of the mainbody**;
> >4. **End for**
> >---
> >5. **For** $t = \frac{1}{1+\Delta_{cs}}T$ **to** $0$ **do**:
> >6. - Compute $z_{t-1}$ from $z_t$ using decoupled cross-attention (**Eqs. (1) and (8) of the mainbody**) of $\epsilon_{\theta}$, conditioned on $C_{te}$, $C_{st}$ following **Eq. (5) of the mainbody**;
> >7. **End for**
> >---
> >8. Output the personalized generated image $I_0 = \mathcal{D}(z_0)$.
>
>
> ---
> # [W7]: The Clarification on the Qualitative Results for Image Customization
>
> **Response**: Thanks for your comments. We kindly remind that as illustrated in **Fig.5(b) of the mainbody**, our DRP-Diff balances both fidelity to the image reference and alignment with the text prompt. For example, methods such as **IP-Adapter and MIP-Adapter and Patchdpo** place more emphasis on fidelity: they can restore the butterfly’s characteristics from the reference image, but **the background regions are generated independently of the text prompts**.
>
> On the other hand, **SSR-Encoder, Dreamcache and   MS-Diffusion** generate images more strictly aligned with the text prompts, yet **the butterfly’s attributes no longer match the reference**. In contrast, **our approach effectively achieves a compromise: it generates images that are well-aligned with the text while largely preserving the key features of the butterfly from the reference image**. Some deviations are inevitable in this process, but the overall balance between fidelity and text alignment is substantially improved. The **Table.1(b) of the mainbody** can also validate that.
>
> ---
> # [W8]: Writing and Presentation
>
> **Response**: Thanks for the comments. We will address all your suggestions regarding writing issues, such as long sentences, small figures, and typos.
>
>
> **_Following your suggestion, we enlarge all the figure size in our refined manuscript version._**
>
> ---
> # [Q1]: Question on the "Sparse Text Prompts"
> **Response**: Thanks. We clarify that we adopt **sparse text prompts** and **dense image references** because images contain much richer information than text prompts. Hence, we refer to text prompts as **sparse** and images as **dense**.
>
> ---
> # [Q2]: Question on How to Disentangle the Background of the Denoised Latent Image Feature
>
> **Response**: Thanks for your valuable comments. We kindly remind that as clearly formulated in **Eq.(7) of the mainbody**, **we copy that as follows:**
>
> - The **decoupled background term** leverages the foreground texture of the style reference to provide complementary information for reconstructing the background of $F$ in **Eq.(7) of the mainbody**, by expanding the key and value sets in the cross-attention layers, thereby facilitating the query $FW_q$ to attend to more regions $C_{st}W_v^{im}$ in **Eq.(7) of the mainbody**, so as to **achieve alignment between the text prompt and the customized reference with a focus on the background**.
>
> - The **auxiliary position term** attends more to background of style reference $C_{st}\Delta W_k^{im}$, thereby **alleviating the inference with the customized reference**.
>
>
> ---
> # Reference
>
> [1] Binyamin L, Tewel Y, Segev H, et al. Make it count: Text-to-image generation with an accurate number of objects[C]//Proceedings of the Computer Vision and Pattern Recognition Conference. 2025: 13242-13251.
>
>
> ---

---

### Official Review · Reviewer_Ukfn · 2025-10-28

**Soundness:** 4
**Presentation:** 3
**Contribution:** 3
**Rating:** 6
**Confidence:** 5

**Summary:**

This paper proposes DRP-Diff, a dual-reference diffusion model for personalized image generation that improves text alignment by disentangling foreground customization and background style. By using separate texture- and style-disentangled matrices, the method aligns background and foreground with text prompts in early and late denoising stages, respectively, guided by an adaptive timestep strategy based on information entropy. Experiments show superior performance over existing approaches.

**Strengths:**

1. The overall writing is easy to understand.
2. The author analyzes the response of principal components in the attention mapping matrix to image foreground and background from a parametric perspective, supported by visualization. Furthermore, a backgroud-decoupled matrix $\Delta W_{k}^{im}$ is introduced to refine the parameters. This design is reasonable and insightful.
3. On basis of the above design, the author modifies the attention interaction, effectively preserving the main semantic content of the subject image while decoupling background information and irrelevant texture details, thus facilitating better integration of texture information from the style image. This is also validated by several experimental results.
4. The author further introduces a background-sensitive matrix $\overline{W}^{im}_k$ in the late stage to help the model better capture low-level features of the style image.

**Weaknesses:**

1. Intuition and Justification for SVD-based Disentanglement: The paper explains the SVD inversion in Equation (5) to enhance background components. While mathematically presented, a deeper intuitive or empirical justification for choosing $\frac{1}{\sigma_i}$ over other potential weighting schemes could strengthen this crucial design choice. Did the authors explore other ways to manipulate singular values (e.g., simply setting the weight of foreground-sensitive principles to 0), and why was this specific form the most effective?
2. Could there be further analysis on the SVD decomposition, such as the numerical distribution of $\sum$ (i.e., the singular values), and whether the parameter matrices across different layers exhibit similar characteristics?
3. This work appears heavily based on IP-Adapter, but it is unclear whether the encoding of reference/style images and attention interaction patterns remain consistent across different customized models. I am concerned about potential over-tuning. Could the authors provide further analysis to validate the generalizability of their method?

**Questions:**

please refer to the weakness

---

> ### Author Response · Authors · 2025-11-22
> **Author Response (part 1 of 2)**
>
> # To Reviewer Ukfn
> Thanks very much for your valuable comments that helped us significantly improve this submission. **We kindly remind that, as per the rebuttal policy, we also submitted our refined manuscript.** Below are detailed point-to-point responses about how we address your questions.
>
>
> ---
> # [Q1]: Question on Foreground and Background Disentanglement via Manipulating Singular Values
>
> **Response**:  Thanks for the constructive comment. Following your suggestions, we further conduct additional ablation studies to investigate different strategies for suppressing foreground components and enhancing background components in the singular values. Specifically, we consider all cases for **the hard suppression approach $\sigma_{\text{top-n\%} = 0}$, which denotes the top n% of singular values to zero, and the soft suppression approach $\sigma e^{-\sigma}$, which represents a reweighting scheme that attenuates larger singular values while preserving smaller ones**, along with the detailed explanations, as elaborated below:
>
> | Case                        | CLIP-T-Mean↑       | CLIP-T-Std↓       | CLIP-I-Mean↑       | CLIP-I-Std↓       |
> |:------------------------------|:-----------------|:----------------|:-----------------|:----------------|
> | $\sigma_{\text{top-10 \%} = 0}$   | 0.2469 (-0.68%)  | 0.0352 (+13.5%)  | **0.6450 (+0.45%)** | 0.0867 (+8.36%) |
> | $\sigma_{\text{top-20\%}=0}$   | 0.2496 (+0.40%)  | 0.0344 (+11.0%)  | 0.6433 (+0.19%)  | 0.0806 (+0.62%) |
> | $\sigma_{\text{top-30\%}=0}$   | **0.2500 (+0.56%)** | 0.0332 (+7.10%) | 0.6401 (-0.31%)  | 0.0842 (+5.11%) |
> | $\sigma_{\text{top-40\%}=0}$   | 0.2495 (+0.36%)  | 0.0334 (+7.74%)  | 0.6382 (-0.30%)  | 0.0826 (+3.11%) |
> | $\sigma_{\text{top-50\%}=0}$   | 0.2499 (+0.52%)  | 0.0322 (+3.87%)  | 0.6416 (-0.08%)  | 0.0868 (+8.36%) |
> | $\sigma e^{-\sigma}$           | 0.2498 (+0.49%)  | 0.0330 (+6.45%)  | 0.6401 (-0.31%)  | 0.0883 (+10.2%) |
> | **Ours ($1 / \sigma$)**            | 0.2486           | **0.0310**       | 0.6421           | **0.0801**       |
> |||
>
> - **Analysis of different strategy to disentangle foreground and background via manipulating singular values**
>
> "*The above table reports the results regarding all cases for singular values: overall, all the adjustment strategies show no significant difference in their ability to extract background information (measured by **CLIP-T and CLIP-I mean scores**); the main variations are observed in the fluctuations across different images (measured by **CLIP-T and CLIP-I std scores**). Our method demonstrates stable background disentanglement across diverse images.*
>
> *1. Setting different proportions of the top 10–50% of singular values to zero does not yield a clear improvement in performance. This is because the hard partitioning of foreground and background components **makes it difficult to adaptively determine the proportion of foreground and background for each image**, leading to limited performance gains and high variability across images.*
>
> *2. The soft suppression approach $\sigma e^{-\sigma}$ fails to effectively disentangle background for each image. While **it adaptively suppresses foreground components, it does not explicitly enhance background components**. As a result, the contribution of background components in the cross-attention remains limited, leading to incomplete background disentanglement.*
>
> *3. Orthogonally, our reciprocal operation $1/\sigma$ **not only weakens the dominant foreground components but also actively amplifies the smaller singular values corresponding to background**. This balanced adjustment ensures both suppression of foreground and enhancement of background, which results in more stable and consistent background disentanglement across images, confirming the intuition in **Sec.2.2 of the mainbody***"
>
> **_Following your comments above, the above results as well as analysis are also made up in Sec. 3.3 of the mainbody for our refined manuscript._**
>
> ---

---

> ### Author Response · Authors · 2025-11-22
> **Author Response (part 2 of 2)**
>
> # [Q2]: Discussion on the Singular Values in the Different Layers
>  As per your suggestion, we further analysis the singular values in the different cross-attention layers based on the singular value decomposition (SVD) on $W_{k}^{im}$ (**Eq.(4) of the mainbody**) and $\overline{W}_{k}^{im}$ (**Eq.(5) of the mainbody**) , along with the detailed explanations.
>
> For the singular value matrices obtained from different layers, we compute **the proportion of the top n% singular values relative to the sum of all singular values**, named $\sum \sigma_{\text{top-n\%}}$. We also take the reciprocal of each singular value and perform the same calculation. In both cases, the proportions are evaluated for $n \in ${2, 5, 10, 20}. The results are reported below:
>
> |Layer Name|Resolution|$\sum \sigma_{\text{top-2\%}}$|$\sum \sigma_{\text{top-5\%}}$|$\sum \sigma_{\text{top-10\%}}$|$\sum \sigma_{\text{top-20\%}}$|$\sum \frac{1}{\sigma}_{\text{top-2\%}}$|$\sum \frac{1}{\sigma}_{\text{top-5\%}}$|$\sum \frac{1}{\sigma}_{\text{top-10\%}}$|$\sum \frac{1}{\sigma}_{\text{top-20\%}}$|
> |:---:|:---:|:---:|:---:|:---:|:---:|:---:|:---:|:---:|:---:|
> |down_blocks.1.0.0|64×64|0.127|0.259|0.408|0.606|0.520|0.724|0.849|0.934|
> |down_blocks.1.1.1|64×64|0.147|0.279|0.431|0.635|0.562|0.756|0.869|0.944|
> |down_blocks.2.0.0|32×32|0.197|0.360|0.544|0.763|0.525|0.715|0.838|0.927|
> |down_blocks.2.1.9|32×32|0.228|0.392|0.562|0.751|0.552|0.733|0.849|0.932|
> |mid_block.0.0|32×32|0.208|0.384|0.573|0.790|0.533|0.721|0.842|0.929|
> |mid_block.0.9|32×32|0.234|0.381|0.536|0.719|0.563|0.739|0.853|0.934|
> |up_blocks.0.0.0|32×32|0.179|0.344|0.532|0.755|0.518|0.710|0.835|0.926|
> |up_blocks.0.2.9|32×32|0.285|0.489|0.652|0.783|0.556|0.735|0.850|0.933|
> |up_blocks.1.0.0|64×64|0.130|0.275|0.437|0.641|0.537|0.739|0.859|0.939|
> |up_blocks.1.2.1|64×64|0.150|0.292|0.448|0.647|0.527|0.733|0.856|0.938|
> |||
>
> - **Discussion on the singular values in the different cross-attention layers**
>
> *The above results indicate that: under the same spatial resolution, the proportions of the top n% singular values exhibit highly consistent patterns across different layers. This consistency supports our claim that **larger $\sigma_i$ tends to correspond to foreground-dominant components, whereas smaller $\sigma_i$ is more closely associated with background information** (**Lines 226-228, Sec. 2.2 of the mainbody**), which confirms universally across layers. Furthermore, after inverting the singular values, the resulting distributions across layers also display similar behaviors, indicating that our style-disentangled matrices $\overline{W}_{k}^{im}$ **effectively amplify the relatively weak background components**, which is in line with **Eq.(5) of the mainbody***"
>
> **_Following your comments above, the above results as well as analysis are also made up in Sec. D.2 of the Appendix for our refined manuscript._**
>
> ---
> # [Q3]: Discussion on the Generalization of DRP-Diff
>
> **Response**: Thanks for your comment. We would like to clarify that, according to the current model architecture, recent works on image customization and style transfer can generally be categorized into two groups:
>
> (1) the methods that require finetuning by converting image inputs into text embeddings, which inevitably leads to a loss of visual information during the conversion process and has therefore **become less commonly adopted in recent works**.
>
> (2) **finetuning-free approaches that modify the decoupled cross-attention mechanism in IP-Adapter**.
>
> Based on this categorization, **our strategy can be seamlessly plugged into all existing finetuning-free personalization frameworks**, as our design directly operates on the decoupled cross-attention structure used in IP-Adapter. This ensures full plug-and-play compatibility within this class of methods.
>
> We also provide qualitative comparisons to validate the generalization. As indicated in **Sec.J of the Appendix**, We report the generated results of several representative text-to-image (T2I) models, including different fine-tuned variants of Stable Diffusion XL (SDXL), namely **Realistic VisionXL** and **DreamshaperXL**, as well as the DiT-based approach **HunyuanDiT**, **we copy that as follows (please also kindly refer to **Fig.10 in the Appendix**)**:
>
> *"These three models exhibit distinct characteristics due to differences in their training data:*
>
> - *__Realistic VisionXL__: Fine-tuned on high-quality, photography-oriented datasets,...,  As shown in Fig.10(a), **the customized content and style achieve a well-balanced integration**, making this model suitable for both image customization and style transfer.*
>
> - *__DreamshaperXL__: Trained with a blend of artistic and stylized datasets,..., As illustrated in Fig.10(b), **its outputs place stronger emphasis on stylization**...*
>
> - *__HunyuanDiT__: Trained on large-scale, multi-domain datasets,..., **this model is more suitable for image customization tasks** within our DRP-Diff framework, as shown in Fig.10(c)."*
>
> ---

---

### Official Review · Reviewer_uCgN · 2025-10-31

**Soundness:** 3
**Presentation:** 2
**Contribution:** 2
**Rating:** 4
**Confidence:** 4

**Summary:**

This paper proposes DRP-Diff, which is a personalization method for T2I diffusion models that works for both image customization (with text condition) and style transfer (with image condition). The paper addresses the text condition & generated image and image condition & generated image misalignment in prior works by disentangling foreground customization and background style. Such disentanglement is enabled by the below two key methods:
1. Customized texture-disentangled matrix and style-disentangled matrix obtained via SVD to separate foreground and background contributions in cross-attention.
2. Early–late denoising stage division, modulated adaptively using an entropy ratio between customized and style references.

Comprehensive experiments show consistent improvements and better visual quality over baselines like IP-Adapter, MS-Diffusion, RB-Modulation, etc.

**Strengths:**

1. The paper's motivation is clear and interesting.
2. The method is, in general, well motivated and technically solid, with most of the details introduced in the paper.
3. The paper's performance is strong across several different metrics and settings.

**Weaknesses:**

1. The paper has several issues considering its writing. (1) The tables, figures, and their fonts are too small, very hard to see. (2) It is sometimes difficult to follow due to excessive formulas and symbols. There could be a notation table for better reference. (3) There appear to be many long sentences. The core intuition could be communicated more concisely.
2. The claim in L204-205 that "larger $\sigma_i$ tend to correspond to the foreground, whereas smaller σi are more likely associated with the background" lacks support from quantitative/statistical supports, only demonstrated by some visualizations.
3. There is lack of analysis and discussions on the method efficiency, i.e., how much more time/compute is added by the method.

**Questions:**

1. L090-L093: The long sentence: "So far, neither image customization nor style transfer fails to well align the generated image with the text
prompts, specifying as misalignment with background for image customization and foreground for style transfer, owing to the dense image references that attend only to either the foreground or background, while overwhelming the sparse text prompts." is very hard to understand, and seem to be in the opposite meaning.
2. Whether the authors can provide quantitative/statistical support for the claim in L204-205?
3. Whether the authors can provide analysis and discussions on the method efficiency, i.e., how much more time/compute is added by the method?

---

> ### Author Response · Authors · 2025-11-22
> **Author Response (part 1 of 2)**
>
> # To Reviewer uCgN
> Thank you very much for your valuable comments that helped us significantly improve this submission, *especially appreciating our motivation and solid technique*. **We kindly remind that, as per the rebuttal policy, we also submitted our refined manuscript**. Below are detailed point-to-point responses about how we address your questions.
>
> ---
> # [Q1]: Writing and Presentation
>
> **Response**: Thanks for the comments. We will address all your suggestions regarding writing issues, such as long sentences, small figures, and typos.
>
> - The original sentence:
>   *"So far, neither image customization nor style transfer fails to well align the generated image with the text prompts, specifying as misalignment with background for image customization and foreground for style transfer, owing to the dense image references that attend only to either the foreground or background, while overwhelming the sparse text prompts."*
>
>   **will be replaced with:**
>   **"_So far, both image customization and style transfer exhibit suboptimal alignment with the text prompts. In particular, image customization primarily suffers from background misalignment, whereas style transfer is prone to foreground misalignment. This misalignment arises from the dense image references that attend only to either the foreground or background, while overwhelming the sparse text prompts._"**
>
> **_Please also refer to Sec.1 of the mainbody in our refined manuscript version._**
>
>
> ---
> # [Q2]: The Quantitative Results for the Intuitions of the Different Singular Values
>
> **Response**: Thanks for the comments. As per your suggestion, to validate the intuition that **larger $\sigma_i$ tends to correspond to the foreground, whereas smaller $\sigma_i$ is more likely associated with the background** (**Lines 226-228, Sec.2.2 of the mainbody**). We further made up more results as shown in Table.1, where we utilize the text prompts for both the foreground and background to generate their corresponding images using a text-to-image model. These generated images are then used as image references in our **DRP-Diff** framework, where we disentangle the foreground via the large singular values and the background via the small singular values, along with the detailed explanations, as shown below:
>
> - Table 1. The examples of text prompts for both foreground and background, foreground, and background
>
> | ID | Both foreground and background | Foreground | Background |
> |:--:|:------------------------------|:-----------|:-----------|
> | 1  | A snow leopard is leaping across a fractured frozen expanse. | a snow leopard | the fractured frozen expanse |
> | 2  | A giant squid is drifting in a radiant deep chasm. | a giant squid | the radiant deep chasm |
> | 3  | A phoenix is glowing over an age-worn conflict field. | the phoenix | the age-worn conflict field |
> | 4  | A red-crowned crane is gliding through a crimson-lit horizon. | a red-crowned crane | the crimson horizon |
> | 5  | A polar bear is sitting beside a fractured icy dome-like form. | a polar bear | the fractured icy dome |
> |||
>
> - **Discussion on the intuitions of the different singular values**
>
> *"For quantitative evaluation, we also compute the similarity between the disentangled images and the text prompts for foreground or background in the Table.2 via the CLIP model. Further, we generate images from text prompts for the foreground or background using a text-to-image model, and then measure the similarity between the disentangled images and these generated images. Both CLIP and DINO models are employed to assess the image-level similarity, providing a comprehensive evaluation of the consistency of the foreground–background disentanglement.*
>
> *The quantitative results summarize our findings below: as illustrated in the left of the Table.2, **the large singular values is able to disentangle more foreground information than the small singular values**, as evidenced by the higher scores in CLIP-T, CLIP-I, and DINO-I, whereas **the small singular values is more effective at disentangling background information**, as shown in the right of the Table.2, confirming the intuition in **Sec.2.2 of the mainbody**"*
>
>
> - Table 2. Quantitative comparisons of foreground–background disentanglement across different images
>
> | Method                 | (Fore) CLIP-T↑ | (Fore) CLIP-I↑ | (Fore) DINO-I↑ | (Back) CLIP-T↑ | (Back)CLIP-I↑ | (Back) DINO-I↑ |
> |:----------------------:|:-----------------:|:-----------------:|:-----------------:|:-----------------:|:-----------------:|:-----------------:|
> |Large singular values| **0.3115**        | **0.8228**        | **0.5774**        | 0.2303            | 0.6157            | 0.1353            |
> |Small singular values| 0.2629            | 0.7090            | 0.2591            | **0.2486**        | **0.6421**        | **0.1966**        |
> |||
>
>
> **_The above results and analysis are also made up in the Sec.D.1 in the Appendix of our revised manuscript._**
>
> ---

---

> ### Author Response · Authors · 2025-11-22
> **Author Response (part 2 of 2)**
>
> # [Q3]: Question on Computational Efficiency
>
> **Response**: Thanks. We mildly remind you that we do present a comparison of computational efficiency with state-of-the-art methods in the **Lines 419-426, Sec. 3.2 of the mainbody, Table.1(a) of the mainbody and the Table.12 of the Appendix, we copy that as follows:**
>
>
> - **Table.1(a) of the mainbody**
>
> | Method                  | Time(s) ↓       | VRAM(GB) ↓       |
> |:-------------------------|:----------------|:----------------|
> | IP-Adapter + StyleSSP   | 9.61           | 4.63           |
> | SSR-Encoder + StyleSSP  | 7.34         | 1.59        |
> | MIP-Adapter + StyleSSP  | 12.16          | 4.53           |
> | Dreamcache + StyleSSP   | 8.48           | **1.51**         |
> | PatchDPO + StyleSSP     | 9.42           | 4.63           |
> | MS-Diffusion + StyleSSP | 10.99          | 4.56           |
> |||
> | **DRP-Diff**              | **5.66 [10.54]**  | 4.55           |
> |||
>
>
> - **The Table.12 of the Appendix**
>
> | Method                  | Time(s) ↓       | VRAM(GB) ↓       |
> |:-------------------------|:----------------|:----------------|
> | IP-Adapter + CSGO       | 17.39          | 4.72           |
> | SSR-Encoder + CSGO      | 15.12        | 1.68        |
> | MIP-Adapter + CSGO      | 19.94          | 4.62           |
> | Dreamcache + CSGO       | 16.26          | 1.60        |
> | PatchDPO + CSGO         | 17.20          | 4.72           |
> | MS-Diffusion + CSGO     | 18.77          | 4.65           |
> |||
> | IP-Adapter + StyleID    | 13.30          | 16.92          |
> | SSR-Encoder + StyleID   | 11.03        | 13.88          |
> | MIP-Adapter + StyleID   | 15.85          | 16.82          |
> | Dreamcache + StyleID    | 12.17          | 13.80        |
> | PatchDPO + StyleID      | 13.11          | 16.92          |
> | MS-Diffusion + StyleID  | 14.68          | 16.85          |
> |||
> | IP-Adapter + StyleSSP   | 9.61           | 4.63           |
> | SSR-Encoder + StyleSSP  | 7.34         | 1.59        |
> | MIP-Adapter + StyleSSP  | 12.16          | 4.53           |
> | Dreamcache + StyleSSP   | 8.48           | **1.51**         |
> | PatchDPO + StyleSSP     | 9.42           | 4.63           |
> | MS-Diffusion + StyleSSP | 10.99          | 4.56           |
> |||
> | **DRP-Diff**              | **5.66 [10.54]**  | 4.55           |
> |||
>
> - **Analysis of the computational efficiency**
>
> _"As clearly indicated by __Lines 419-426, Sec. 3.2 of the mainbody__: inference time of our DRP-Diff with two parts: 5.66s for the denoising process and 4.88s for computing **Eq.(5) and Eq.(6)**. This demonstrates superior efficiency compared with state-of-the-art methods. Moreover, since **Eq.(5) and Eq.(6) are parameter-dependent and unrelated to the input data, they can be pre-computed during the model loading stage, thereby saving this computation cost during inference**. Our DRP-Diff also achieves competitive results in VRAM consumption. Specifically, SSR-Encoder adopts SD v1.5 and Dreamcache is designed for lightweight usage, so both consume less VRAM. However, compared with other SDXL-based models, our **DRP-Diff achieves the lowest VRAM usage** because it completes both tasks within a single denoising process."_
>
> - As per your suggestion, we extend the experiments in discussion to report **the additional inference time and memory usage** introduced by our method, along with the detailed explanations, as shown below:
>
>
> | Method                  | Time(s) ↓       | VRAM(GB) ↓       |
> |:-------------------------|:----------------|:----------------|
> | IP-Adapter       | **4.75**          | **4.00**           |
> | **DRP-Diff**              | 5.66 | 4.55           |
> |||
>
> *"Compared to the IP-Adapter using a single image reference, our DRP-Diff with two image references bears only additional 0.91s in inference time and 0.55GB in peak VRAM usage, owing to the fact that **Eq.(5) and Eq.(6) of the mainbody** are parameter-dependent and unrelated to the input data, they can be pre-computed during the model loading stage, thereby saving this computation cost during inference, which is in line with  __Sec.2.2 of the mainbody__."*
>
>
> **_The above results and analysis are also made up in the Sec.I in the Appendix of our revised manuscript._**
>
> ---

---

### Official Review · Reviewer_GChA · 2025-10-31

**Soundness:** 3
**Presentation:** 3
**Contribution:** 3
**Rating:** 6
**Confidence:** 3

**Summary:**

This paper introduces DRP-Diff, a dual-reference personalized diffusion framework that addresses text-image misalignment in customized image generation. It identifies that existing methods overemphasize either the foreground or background, leading to mismatched content. DRP-Diff disentangles style and texture via SVD-based matrices and employs stage-specific processing: early stages align backgrounds using dual-reference textures, while late stages enhance text-consistent foregrounds through adaptive timestep modulation guided by information entropy. Extensive experiments on DreamBench and StyleBench show superior performance over CLIP-based baselines. The framework offers both practical improvements and theoretical insights into diffusion model behavior in personalized generation.

**Strengths:**

1. Systematically leverages the diffusion process’s global-to-local progression to address text alignment, introducing style disentanglement, texture disentanglement, and adaptive timestep modulation.

2. Comprehensive ablations confirm each module’s contribution to detail preservation, style quality, and entropy-based adaptivity.

3. Provides full mathematical derivations, implementation details, and open-source code for reproducibility and theoretical soundness.

**Weaknesses:**

While the paper claims that "the high-level noise in the early stage benefits background reconstruction" and "the late stage prefers foreground reconstruction," it lacks rigorous theoretical analysis connecting diffusion process characteristics to text alignment. The paper cites prior work (Wu et al., 2023; Chen et al., 2023) that observed semantic hierarchy in diffusion models, but fails to explain why this hierarchy specifically addresses the text alignment problem. For instance, Section 2.2 mentions "owing to the high-level noise in the early stage to benefit the background reconstruction," although it may seem right, but provides no mathematical derivation or frequency analysis to support this claim.

**Questions:**

1. The paper demonstrates the effectiveness of the adaptive timestep modulation mechanism through Figure 6 and Case F in Table 2(b). However, could the authors provide a more detailed analysis on the specific behaviors and handling strategies when H_cu and H_st exhibit extreme deviations?
﻿
2. The paper mentions generating C^te_bg by replacing the foreground in C^te with {}, but does not explain how to automatically identify the foreground components within the text prompts. For complex prompts such as "The glowing moon rose above the distant hill while a group of birds flew across the calm sea," how does the system accurately distinguish multiple foreground objects from the background? Could the authors provide an error rate analysis and specify the exact entity recognition methodology employed?

3. Figure 7 illustrates that StyleStudio encounters difficulties with multi-object prompts (e.g., "incorrect number of 'woman'"). However, the paper does not elaborate on how DRP-Diff specifically addresses this issue. How does the method's performance change as the number of objects increases (e.g., "seven women") or when inter-object relationships become more complex? Could the authors provide a quantitative relationship analysis between object count and performance metrics?

---

> ### Author Response · Authors · 2025-11-22
> **Author Response (part 1 of 3)**
>
> # To Reviewer GChA
> Thank you very much for your valuable comments that helped us significantly improve this submission. **We kindly remind that, as per the rebuttal policy, we also submitted our refined manuscript**. Below are detailed point-to-point responses about how we address your questions.
>
> ---
> # [W1]: The Clarification on the Relation between Denoising Process and Text prompts.
> **Response**: Thanks for your valuable comments. We kindly clarify the claims from **Lines 257-260, Sec.2.2 of the mainbody**, that **during the denoising process, images recover low-frequency band, followed by high-frequency details**. **Text prompt primarily corresponds to low-frequency band**, so in the early denoising stage with high-level noise, it is most effective for generating the overall content and background structure. **Foreground is already guided by the image reference, so text prompt mainly contributes to generating the background, ensuring semantic alignment with the text prompt**. In the late stage, as the noise level decreases, while the foreground and background are largely established, style references can be used to refine foreground details, hence improving high-frequency details.
>
> Following your suggestions on validating the claims via frequency analysis, we experimentally validate the denoising process as per several metrics to measure changes regarding the low-and-high frequency bands at different timesteps during the denoising process, as reported below:
>
> | Timesteps | Mean Gray | Mean HSV-V | Mean Luma | Avg Gradient | Variance | LBP Variance |
> |:-----------|:---------|:----------|:---------|:------------|:--------|:------------|
> |**0**|**120.9**|**136.0**|**120.9**|**59.01**|**3955.0**|**4.45**|
> |5|120.8|134.6|120.8|75.07|4098.3|3.71|
> |10|120.7|133.5|120.7|101.58|4378.9|3.25|
> |15|120.7|133.7|120.7|133.26|4790.8|3.07|
> |20|120.9|135.1|120.9|164.23|5274.8|3.07|
> |25|121.2|137.3|121.2|190.49|5743.9|3.15|
> |30|121.4|139.2|121.4|209.84|6099.5|3.25|
> |35|121.1|140.0|121.1|222.03|6291.1|3.32|
> |40|120.5|140.1|120.5|228.75|6363.9|3.35|
> |45|119.9|139.8|119.9|232.19|6394.8|3.38|
> |50|119.5|139.5|119.5|233.88|6419.5|3.40|
> |||
>
> * **Analysis of the high-and-low frequency in the denoising process**
>
> *For **low-frequency information**, we adopt **Mean Gray**, **Mean HSV-V**, and **Mean Luma**, which primarily capture the global brightness and luminance structure of the denoised image. These metrics help evaluate how well the low-frequency bands are maintained throughout the denoising process.*
>
> *For **high-frequency information**, we use **Average Gradient**, **Variance**, and **LBP Variance**, which are sensitive to edge sharpness, local contrast, and texture complexity, respectively. These metrics reflect the preservation and recovery of fine details during denoising.*
>
> *The results show that low-frequency metrics (Mean Gray, HSV-V, Luma) remain nearly unchanged from the beginning (i.e., **the Mean Gray value only shifts slightly from 119.5 to 120.9**). In contrast, high-frequency metrics (Avg Gradient, Variance, LBP Variance) increase steadily (i.e., **Avg Gradient decreases from 233.88 to 59.01**) in the late timesteps, confirming that high-frequency details are refined gradually during denoising, which is in line with __Lines 257-260, Sec.2.2 of the mainbody__".*
>
>
> **_The above results and analysis are also made up in Sec.C of the Appendix of our revised manuscript._**
>
>
> ---

---

> ### Author Response · Authors · 2025-11-22
> **Author Response (part 2 of 3)**
>
> # [Q1]: Question on Adaptively Modulating the Early and Late Stages
> **Response**: Thanks for your valuable comments. We claim the $H_{cu}$ and $H_{st}$ in  **Eq.(9) of the mainbody**, where the intuition is that, **unlike the sparsity of text prompts, the differences in information entropy among images are generally modest**. Therefore, the threshold ($\Delta_{cs}$) tends to remain within a stable range rather than dramatic variation.
>
> To validate the above, we further conduct the ablation studies on exploiting denoising timesteps between the early and late stages by dividing the denoising process into early and late stages according to vary the proportion of the early stage, *i.e.*, \{0.1; 0.2; 0.3; 0.4; 0.5\}, We also provide the result of adaptively modulating the early and late denoising stage (**Eq.(9) of the mainbody**), along with the detailed explanations, as shown below:
>
> | Method  | CLIP-T↑   | CLIP-C↑   | CLIP-S↑   | DINO-C ↑  | DINO-S↑   | Mean(CLIP)↑   | Mean(DINO) ↑  |
> | :-----: | :----: | :----: | :----: | :----: | :----: | :--------: | :--------: |
> | 0.1       | 0.2864 | 0.6631 | **0.7036** | 0.3098 | **0.4459** | 0.5510     | 0.37785    |
> | 0.2       | 0.2954 | 0.6963 | 0.6821 | 0.3830 | 0.4139 | 0.5579     | 0.39845    |
> | 0.3       | 0.3035 | 0.7295 | 0.6558 | 0.4515 | 0.3682 | 0.5629     | 0.40985    |
> | 0.4       | 0.3105 | 0.7554 | 0.6313 | 0.5176 | 0.3237 | 0.5657     | 0.42065    |
> | 0.5       | **0.3166** | **0.7822** | 0.6104 | **0.5705** | 0.2839 | 0.5697     | 0.4262     |
> | **DRP-Diff** | 0.3164 | 0.7764 | 0.6187 | 0.5609 | 0.2958 | **0.5705**  | **0.42835**  |
> |||
>
> * **Discussion on Denoising Timesteps Between the Early and Late Stages.**
>
> "*The results show that when the proportion is set to 0.1, the generated images show the highest similarity to the style reference (**CLIP-S** and **DINO-S**) while being least similar to the customized reference (**CLIP-T**, **CLIP-C** and **DINO-C**) ; conversely, at 0.5, the opposite observation holds. We can also conclude that **DRP-Diff** lies between the results of the 0.4 and 0.5 proportions, demonstrating that **our adaptive modulation strategy can effectively regulate the ratio of early to late stages, leading to a well-balanced performance in terms of both text alignment and dual-image similarity** (**Mean(DINO)** and **Mean(CLIP)** ), confirming the intuition of __Sec.2.4 of the mainbody__.*"
>
> **_The above results and analysis are also made up in Sec.E of the Appendix of our revised manuscript._**
>
> ---
> # [Q2]: Question on How to Identify the Foreground and Background
> **Response**: Thanks. We mildly clarify the **Eqs.(1) and (7)** for identification of the foreground and background: the decoupled cross-attention of the text prompt (**Eq.(1) of the mainbody**) employs the text prompts $C_{te}$ describing **both foreground and background** to generate the images, while in the decoupled cross-attention of the image (**Eq.(7) of the mainbody**):
>
> - For the **foreground** of the denoised personalized image, since **Eq.(7)** indicates that **the customized reference $C_{cu}$ and the text prompts corresponding to the foreground in $C_{te}$ share the same semantics**, the attention score map between the query foreground and $C_{cu}$ can accurately localize the regions described in the text prompts. This is further validated in **the third row of Fig. 1(a) in the main body**, where the text prompt "dog" aligns precisely with the attention region corresponding to the "dog" in the reference image.
>
> - For the **background** of the denoised personalized image, as noted in **Lines 290-296, Sec. 2.3 of the main body**, the background-specific prompts $\overline{C}_{te}$ in **Eq.(7)** serves as an **auxiliary position term** that attends more to the background, thereby mitigating interference from the customized reference. To further validate this, we conduct ablation studies reported in **Lines 486-494, Sec. 3.3 of the main body, i.e., Table. 3, we copy that below:**
>
> | Method|  CLIP-T ↑| CLIP-TC ↑| Avg(CLIP-T, TC)↑ |
> |:------------|:--------------|:---------------|:-----------------|
> | Case C       | 0.3179 (↓5.44%) | 0.5732 (↓0.17%) | 0.4456 (↓2.11%) |
> | **DRP-Diff** | **0.3362**     | **0.5742**      | **0.4552**       |
> |||
>
> "*...Case C: removing the auxiliary term in __Eq.(7)__. Table.3(a) reports the results for all three cases, Our DRP-Diff maintains achieves the large performance gain over Case C implying the auxiliary term tends to alleviate the interference of the style reference with the customized reference, as discussed in __Sec.2.3 of the mainbody__.*"
>
> ---

---

> ### Author Response · Authors · 2025-11-22
> **Author Response (part 3 of 3)**
>
> # [Q3]: Question on the Style Transfer
>
> **Response**: Thanks for the valuable comments. We mildly clarify that we degrade **DRP-Diff** to a setting using only a single style reference for style transfer. As mentioned in **Sec.1 of the mainbody**, the **_dense background-oriented style information leaking into the foreground, resulting in misalignment with the text prompts_**. To address this issue, in the early denoising stage, we leverage the foreground texture of the style reference through the customized texture-disentangled matrix $\Delta W_k^{im}$ (**Eq.(6) of the mainbody**) to reconstruct the foreground of the denoised personalized image. In contrast, during the late denoising stage, we utilize the background of the style reference via the style-disentangled matrices $\overline{W}_k^{im}$ (**Eq.(5) of the mainbody**), which act as the key together with their corresponding value.
>
> * **About complexness for inter-object relationship**
>
> As clearly mentioned in **Lines 965-969, Sec. F of the Appendix**, we utilize **T2I-CompBench++** to provide the text prompts for stylized image generation. T2I-CompBench++ comprises **8,000** compositional text prompts categorized into 4 primary groups: attribute binding, object relationships, generative numeracy, and complex compositions. Notably, **each category contains text prompts distributed across varying levels of complexity**; e.g., the numeracy category includes simple prompts such as "two boys" as well as more challenging ones such as "two pigs, four shrimp and a cake."
>
> Following your suggestion, we made up the quantitative relationship between object count and performance metrics, along with the detailed explanations, as shown below:
>
> | Difficulty|CLIP-TS↑  |CLIP-S↑  | DINO-S↑  |
> |:--------------------------|:------|:------|:------|
> |Easy| **0.4966** | **0.7310** | **0.5313** |
> |Medium | 0.4954 | 0.7261 | 0.5273 |
> |Hard | 0.4951 | 0.7236 | 0.5263 |
> |||
>
> * **Analysis of the quantitative relationship between object count and performance metrics**
>
> "*We conduct more ablation studies by reducing the number of objects described in the text prompts from the numeracy category of T2I-CompBench++, thereby forming prompts of different difficulty levels: __Easy (0–3 objects), Medium (4–6 objects), and Hard (more than 6 objects)__. The above quantitative results show that as the text prompts increase in complexity—from Easy to Medium to Hard—the generation task becomes inherently more difficult. __Text prompts contain more objects impose stronger compositional constraints on the model, making it progressively harder to maintain consistency with both the text prompts and the style reference__. As a result, performance across the metrics is progressively reduced, which indicates that DRP-Diff outperforms by disentangle the foreground and background references (__Sec. 1 of the mainbody__).*"
>
> **_The above results and analysis are also made up in Sec.H of the Appendix of our revised manuscript._**
>
> ---

---

### Meta-Review · Area_Chair_NVZZ · 2026-01-07

**Summary:**

This paper proposes DRP-Diff, a dual-reference diffusion framework for personalized image generation that jointly addresses image customization and style transfer by improving text–image alignment. The key insight is to disentangle foreground customization and background style, which are shown to interfere with text prompts in prior single-reference methods. The method reformulates cross-attention using SVD-based texture- and style-disentangled matrices, and applies them at different denoising stages, leveraging the global-to-local behavior of diffusion models. An adaptive early–late timestep modulation strategy based on entropy ratios determines how foreground and background information are injected during denoising. Extensive experiments on DreamBench, StyleBench, and compositional benchmarks demonstrate improved text alignment, image fidelity, and efficiency over state-of-the-art personalization baselines.

**Reviewer Concerns:**

Across all reviewers, the main concerns are:

Justification and theoretical grounding: Several reviewers questioned whether the assumed correspondence between early/late denoising stages and background/foreground reconstruction is sufficiently justified theoretically, noting that the argument relies mainly on intuition and empirical observation rather than a formal diffusion analysis.

SVD-based disentanglement rationale: While most reviewers found the SVD formulation interesting, some felt the choice of manipulating singular values (especially reciprocal weighting) lacked strong intuitive or comparative justification relative to simpler alternatives, and initially lacked quantitative support.

Writing clarity and presentation quality: Multiple reviewers criticized the paper for dense notation, long sentences, small figures/tables, and overall readability issues, which made the method harder to follow despite its technical merit.

Foreground/background definition and text parsing: Reviewers raised concerns about how foreground and background entities are identified from complex text prompts, particularly for multi-object and compositional descriptions, and whether errors propagate from this step.

**Reviewer Scores:**

The scores are 6,4,6,2. None of the reviewers mentions about increasing scores during discussion.

---

### Decision · Program_Chairs · 2026-01-26

Reject